# How Does Message Passing Improve Collaborative Filtering?

**Clark Mingxuan Ju**[1,2*]**, William Shiao**[3]**, Zhichun Guo**[2]**, Yanfang Ye**[2]**,**
**Yozen Liu**[1]**, Neil Shah**[1]**, Tong Zhao**[1*]
[1]Snap Inc., [2]University of Notre Dame, [3]University of California, Riverside
[1]{mju,yliu2,nshah,tong}@snap.com, [2]{mju2,zguo5,yye7}@nd.edu, [3]wshia002@ucr.edu

## Abstract

Collaborative filtering (CF) has exhibited prominent results for recommender systems and been broadly utilized for real-world applications. A branch of research enhances CF methods by message passing (MP) used in graph neural networks, due to its strong capabilities of extracting knowledge from graph-structured data, like user-item bipartite graphs that naturally exist in CF. They assume that MP helps CF methods in a manner akin to its benefits for graph-based learning tasks in general (e.g., node classification). However, even though MP empirically improves CF, whether or not this assumption is correct still needs verification. To address this gap, we formally investigate why MP helps CF from multiple perspectives and show that many assumptions made by previous works are not entirely accurate. With our curated ablation studies and theoretical analyses, we discover that *(i) MP improves the CF performance primarily by additional representations passed from neighbors during the forward pass instead of additional gradient updates to neighbor representations during the model back-propagation and (ii) MP usually helps low-degree nodes more than high-degree nodes.* Utilizing these novel findings, we present **T**est-time **Ag**gregation for **C**ollaborative **F**iltering , namely **TAG-CF**, a test-time augmentation framework that only conducts MP once at inference time. The key novelty of TAG-CF is that it effectively utilizes graph knowledge while circumventing most of notorious computational overheads of MP. Besides, TAG-CF is extremely versatile can be used as a plug-and-play module to enhance representations trained by different CF supervision signals. Evaluated on six datasets (i.e., five academic benchmarks and one real-world industrial dataset), TAG-CF consistently improves the recommendation performance of CF methods without graph by up to **39.2%** on cold users and **31.7**% on all users, with little to no extra computational overheads. Furthermore, compared with trending graph-enhanced CF methods, TAG-CF delivers comparable or even better performance *with less than **1%** of their total training times*. Our code is publicly available at https://github.com/snap-research/Test-time-Aggregation-for-CF.

## 1 Introduction

Recommender systems are essential in improving users' experiences on web services, such as product recommendations [59, 47], video recommendations [15, 55], friend suggestions [46, 72], etc. In particular, recommender systems based on collaborative filtering (CF) have shown superior performance [45, 33, 4]. CF methods use preferences for items by users to predict additional topics

---

*Corresponding authors. Work done during first-author's internship at Snap Inc..

38th Conference on Neural Information Processing Systems (NeurIPS 2024).

or products a user might like [51]. These methods typically learn a unique representation for each user/item and an item is recommended to a user according to their representation similarities [21, 58].

One popular line of research explores Graph Neural Networks (GNNs) for CF, exhibiting improved results compared with CF frameworks without the utilization of graphs [66, 20, 60, 74, 1, 61, 80]. The key mechanism behind GNNs is message passing (MP), where each node aggregates information from its neighbors in the graph, and information from neighbors that are multiple hops away can be acquired by stacked MP layers [30, 56, 18]. During the model training, traditional CF methods directly fetch user/item representations of an observed interaction (e.g., purchase, friending, click, etc.) and enforce their pair-wise similarity [45]. Graph-enhanced CF methods extend this scheme by conducting stacked MP layers over the user-item bipartite graph, and harnessing the resulting user and item representations to calculate a pair-wise affinity.

A recent study [20] shows that removing several key components of the MP layer (e.g., learnable transformation parameters) greatly enhances GNNs' performance for CF. Its proposed method (LightGCN) achieves promising performance by linearly aggregating neighbor representations and has been used as the de facto backbone model for later works due to its simple and effective design [1, 74, 65]. However, this observation contradicts GNN architectures for classic graph learning tasks, where GNNs without these components severely under-perform [41, 62]. Additionally, existing research [20, 60] assumes that the contribution of MP for CF is similar to that for graph learning tasks in general (e.g., node classification or link prediction) - they posit that node representations are progressively refined by their neighbor information and the performance gain is positively proportional to the neighborhood density as measured in node degrees [54]. However, according to our empirical studies in Section 3.2, MP in CF improves low-degree nodes more than high-degree nodes, which also contradicts GNNs' behaviors for classic tasks [54, 23]. In light of these inconsistencies, we ask:

### *What role does message passing really play for collaborative filtering?*

In this work, we investigate contributions brought by MP for CF from two perspectives. Firstly, we unroll the formulation of MP layer and show that its performance improvement could either come from additional representations passed from neighbors during the forward pass or accompanying gradient updates to neighbor representations during the back-propagation. With rigorously designed ablation studies, we empirically demonstrate that gains brought by the forward pass dominate those by the back-propagation. Furthermore, we analyze the performance distribution w.r.t. the user degree (i.e., the number of interactions per user) with or without message passing and discover that the message passing in CF improves low-degree users more compared to high-degree users. For the first time, we connect this phenomenon to Laplacian matrix learning [82, 10, 9], and theoretically show that popular supervision signals [45, 57] for CF inadvertently conduct message passing in the back-propagation even without treating the input data as a graph. Hence, when message passing is applied, high-degree users demonstrate limited improvement, as the benefit of message passing for high degree nodes has already been captured by the supervision signal.

With the above takeaways, we present **T**est-time **Ag**gregation for **C**ollaborative **F**iltering , namely **TAG-CF**. Specifically, unlike other graph CF methods, TAG-CF *does not require any message passing during training*. Instead, it is a test-time augmentation framework that only conducts a *single message-passing step at inference time*, and effectively enhances representations inferred from different CF supervision signals. The test-time design is inspired by our first perspective that, within total performance gains brought by message passing, gains from the forward pass dominate those brought by the backward pass. Applying message passing only at test time avoids repetitive queries (i.e., once per node and epoch) for representations of surrounding nodes, which grow exponentially as the number of layers increases. Moreover, following our second perspective that message passing helps low-degree nodes more in CF, we further offload the cost of TAG-CF by applying the one-time message passing only to low-degree nodes. We summarize our contributions as:

- This is the first work that formally investigates why message passing helps collaborative filtering. We demonstrate that message passing in CF improves the performance primarily by additional representations passed from neighbors during the forward pass instead of accompanying gradient updates to neighbors during the back-propagation, and prove that message passing helps low-degree nodes more than high-degree nodes.

- Given our findings, we propose TAG-CF, an efficient yet effective test-time aggregation framework to enhance representations inferred by different CF supervision signals such as BPR and DirectAU.

Evaluated on six datasets, TAG-CF consistently improves the performance of CF methods without graph by up to **39.2%** on cold users and **31.7%** on all users, with little to no extra computational overheads. Furthermore, compared with trending graph-enhanced CF methods, TAG-CF delivers comparable or even better performance *with less than **1%** of their total training time*.

- Beside promising cost-effectiveness, we show that test-time aggregation in TAG-CF improves the recommendation performance in similar ways as the training-time aggregation does, further demonstrating the legitimacy of our findings.

## 2 Preliminaries and Related Work

**Collaborative Filtering**. Given a set of users, a set of items, and interactions between users and items, collaborative filtering (CF) methods aim at learning a unique representation for each user and item, such that user and item representations can reconstruct all observable interactions [45, 57, 32]. CF methods based on matrix factorization directly utilize the inner product between a pair of user and item representations to infer the existence of their interaction [32, 45]. CF methods based on neural predictors use multi-layer feed-forward neural networks that take user and item representations as inputs and output prediction results [21, 77]. Let $\mathcal{U}$ and $\mathcal{I}$ denote the user set and item set respectively, with user $u_i \in \mathcal{U}$ associated with an embedding $\mathbf{u}_i \in \mathbb{R}^d$ and item $i_i \in \mathcal{U}$ associated with $\mathbf{i}_i \in \mathbb{R}^d$, the similarity $s_{ij}$ between user $u_i$ and item $i_j$ is formulated as $s_{ij} = \hat{\mathbf{u}}_i^\intercal \cdot \hat{\mathbf{i}}_j$.

**Graph Neural Networks**. Graph neural networks (GNNs) are powerful learning frameworks to extract representative information from graphs [30, 56, 18, 70, 11, 26], with numerous applications in large-scale ranking and forecasting tasks [53, 52, 8, 49]. They aim to map each input node into low-dimensional vectors, which can be utilized to conduct either graph-level [69] or node-level tasks [30]. Most GNNs explore layer-wise message passing [14], where each node iteratively extracts information from its first-order neighbors, and information from multi-hop neighbors can be captured by stacked layers. Given a graph $\mathcal{G} = (\mathcal{V}, \mathcal{E})$ and node features $\mathbf{X} \in \mathbb{R}^{|\mathcal{V}| \times d}$, graph convolution in GCN [30] at $k$-th layer is formulated as:

$$\mathbf{h}_i^{(k+1)} = \sigma\left( \sum_{j \in \mathcal{N}(i) \cup i} \frac{1}{\sqrt{|N(i)|}\sqrt{|N(j)|}} \mathbf{h}_j^{(k)} \cdot \mathbf{W}^{(k)} \right), \tag{1}$$

where $\mathbf{h}_i^0 = \mathbf{x}_i$, $\mathcal{N}(i)$ refers to the set of direct neighbors of node $i$, and $\mathbf{W}^{(k)} \in \mathbb{R}^{d^k \times d^{(k+1)}}$ refers to parameters at the $k$-th layer transforming the node representation from $d^k$ to $d^{(k+1)}$ dimension.

Recent works [37, 38] have shown that GNNs make predictions based on the distribution of node neighborhoods. Moreover, GNNs' performance improvement for high-degree nodes is typically better than for low-degree nodes [54, 23, 28, 17, 63]. They posit that node representations are progressively refined by their neighbor information and the performance gain is positively proportional to the neighborhood density as measured in node degrees. As we explore test-time augmentation in this work, it is worth noting that there also exist a group of relevant works that explore data augmentation techniques to enhance the GNN performance [28, 79, 78, 25, 24].

**Message Passing for Collaborative Filtering**. Recent research tends to apply the message passing scheme in GNNs to CF [20, 60, 42, 12, 50, 68, 31, 34]. In CF, they mostly conduct message passing between user-item bipartite graphs and utilize the resultant representations to calculate user-item similarities. For instance, NGCF [60] directly migrates the message passing scheme in GNNs (similar to Equation (1)) and applies it to bipartite graphs in CF. LightGCN [20] simplifies NGCF [60] by removing certain components (i.e., the self-loop, learning parameters for graph convolution, and activation functions) and further improves the recommendation performance compared with NGCF. The simplified parameter-less message passing in LightGCN can be expressed as:

$$\mathbf{u}_i^{(k)} = \sum_{i_j \in N(u_i)} \frac{1}{\sqrt{|N(u_i)|}\sqrt{|N(i_j)|}} \mathbf{i}_j^{(k-1)}, \mathbf{i}_i^{(k)} = \sum_{u_j \in N(i_i)} \frac{1}{\sqrt{|N(i_i)|}\sqrt{|N(u_j)|}} \mathbf{u}_j^{(k-1)}, \tag{2}$$

where $N(\cdot)$ refers to the set of items or users that the input interacts with, $\mathbf{u}_i^{(0)} = \mathbf{u}_i$, and $\mathbf{i}_i^{(0)} = \mathbf{i}_i$. With $K$ layers, the final user/item representations and their similarities are constructed as:

$$\hat{\mathbf{u}}_i = \frac{1}{K+1} \sum_{k=0}^{K} \mathbf{u}_i^{(k)}, \quad \hat{\mathbf{i}}_i = \frac{1}{K+1} \sum_{k=0}^{K} \mathbf{i}_i^{(k)}, \quad s_{ij} = \hat{\mathbf{u}}_i^\intercal \cdot \hat{\mathbf{i}}_j. \tag{3}$$

According to results reported in LightGCN and NGCF [20, 60, 2, 13] and empirical studies we provide in this work (i.e., Table 2 and Table 5), incorporating message passing to CF methods without graphs (i.e., matrix factorization methods [45, 21]) can improve the recommendation performance by up to 20%. Utilizing LightGCN as the backbone model, later works try to further improve the performance by incorporating self-supervised learning signals [35, 74, 1, 73, 64, 27]. Graph-based CF methods assume that the contribution of message passing for CF is similar to that for graph learning tasks in general (e.g., node classification or link prediction). However, whether or not this assumption is correct still needs verification, even though message passing empirically improves CF. There also exists a branch of research that aims at accelerating or simplifying message passing in CF by adding graph-based regularization terms during the training [48, 39, 44, 67]. While promising, they still repetitively query representations of adjacent nodes during the training.

**Efficient Efforts in Matrix Factorization**. A branch of research specifically focuses on improving the efficiency of matrix factorization [48, 44, 22, 43, 6]. For instance, GFCF [48] and Turbo-CF [43] explore graph signal processing to linearly convolve the interaction matrix and use the resulted matrix directly for recommendation without training. Furthermore, SVD-GCN [44] and SVD-AE [22] utilize a low rank version of the interaction matrix to further accelerate the convolution efficiency and yet remain the promising performance. Besides, BSPM [6] studies using diffusion process to gradually reconstruct the interaction matrix and achieves promising performance with fast processing. In parallel with these existing efforts, we propose to enhance any existing matrix factorization method through test-time augmentation that harnesses graph-based heuristics.

## 3 How Does Message Passing Improve Collaborative Filtering?

In this section, we demonstrate why message passing (MP) helps collaborative filtering from two major perspectives: Firstly, we focus on inductive biases brought by the MP explored in LightGCN, the de facto backbone model for graph-based CF methods. Secondly, we consider the performance improvement on different node subgroups w.r.t. the node degree with and without MP.

### 3.1 Neighbor Information vs. Accompanying Gradients from Message Passing

Following the definition in Equation (2), given a one-layer LightGCN[2], we unroll the calculation of the similarity $s_{ij}$ between any user $u_i$ and item $i_j$ as the following:

$$
\begin{aligned}
s_{ij} &= \left( \mathbf{u}_i + \sum_{i_n \in N(u_i)} \frac{1}{\sqrt{|N(u_i)|}\sqrt{|N(i_n)|}} \mathbf{i}_n \right)^\mathsf{T} \cdot \left( \mathbf{i}_j + \sum_{u_n \in N(i_j)} \frac{1}{\sqrt{|N(i_j)|}\sqrt{|N(u_n)|}} \mathbf{u}_n \right) \\
&= \mathbf{u}_i^\mathsf{T} \cdot \mathbf{i}_j + \sum_{u_n \in N(i_j)} \frac{1}{\sqrt{|N(i_j)|}\sqrt{|N(u_n)|}} \mathbf{u}_i^\mathsf{T} \cdot \mathbf{u}_n \sum_{i_n \in N(u_i)} \frac{1}{\sqrt{|N(u_i)|}\sqrt{|N(i_n)|}} \mathbf{i}_n^\mathsf{T} \cdot \mathbf{i}_j \\
&\quad + \sum_{i_n \in N(u_i)} \sum_{u_n \in N(i_j)} \frac{1}{\sqrt{|N(u_i)|}\sqrt{|N(i_n)|}\sqrt{|N(i_j)|}\sqrt{|N(u_n)|}} \mathbf{i}_n^\mathsf{T} \cdot \mathbf{u}_n.
\end{aligned}
\tag{4}
$$

With derived similarities between user-item pairs, their corresponding representations can be updated by objectives (e.g., BPR [45] and DirectAU [57]) that enforce the pair-wise similarity between representations of user-item pairs in the training data.

CF methods without the utilization of graphs directly calculate the similarity between a user and an item with their own representations (i.e., $s_{ij} = \mathbf{u}_i^\mathsf{T} \cdot \mathbf{i}_j$), which aligns with the first term in Equation (4). Compared to the formulation in Equation (4), we can see that three additional similarity terms are introduced as inductive biases: similarities between users who purchase the same item (i.e., $\mathbf{u}_i^\mathsf{T} \cdot \mathbf{u}_n$), between items that share the same buyer (i.e., $\mathbf{i}_n^\mathsf{T} \cdot \mathbf{i}_j$), and between neighbors of an observed interaction (i.e., $\mathbf{i}_n^\mathsf{T} \cdot \mathbf{u}_n$). With these three additional terms from MP, we reason that the performance improvement brought by MP to CF methods without graph could come from **(i)** additional neighbor representations during the forward pass (i.e., numerical values of three extra terms in Equation (4)), or **(ii)** accompanying gradient updates to neighbors during the back-propagation.

---

[2]For the simplicity of the notation, we showcase our observation with only one layer. However, since LightGCN is fully linear, the phenomenon we show also applies to variants with arbitrary layers.

To investigate the origin of the performance improvement brought by MP, we designed two variants of LightGCN. The first one (LightGCN$_{\text{w/o neigh. info}}$) shares the same forward and backward procedures as LightGCN during the training but does not conduct MP during the test time. In this variant, additional gradients brought by MP are maintained as part of the resulting model, but information from neighbors are ablated. In the second variant (LightGCN$_{\text{w/o grad.}}$), the model shares the same forward pass but drops gradients from these three additional terms during the backward propagation. Besides these two variants, we also experiment on LightGCN without MP, denoted as LightGCN$_{\text{w/o both}}$, a matrix factorization model with the same supervision signal (i.e., BPR loss). Implementation details w.r.t. this experiment are in Appendix C.

From Table 1, we observe that the performance of all variants is downgraded compared with LightGCN, with the most significant degradation on LightGCN$_{\text{w/o neigh. info}}$. This phenomenon indicates that **(i)** both additional representations passed from neighbors during the forward pass and accompanying gradient updates to neighbors during the back-propagation help the recommendation performance, and **(ii)** within total performance gains brought by MP, gains from the forward pass dominate those brought

Table 1: Performance of LightGCN variants.

| Method | Yelp-2018 | Gowalla | Amazon-book |
|---|---|---|---|
| | NDCG@20 | | |
| LightGCN | 6.36 | 9.88 | 8.13 |
| w/o grad. | 6.16 (3.1%↓) | 9.87 (0.1%↓) | 7.80 (4.1%↓) |
| w/o neigh. info | 4.71 (25.9%↓) | 6.95 (29.7%↓) | 6.95 (14.5%↓) |
| w/o both | 6.09 (4.2%↓) | 9.83 (0.5%↓) | 7.75 (4.7%↓) |
| | Recall@20 | | |
| LightGCN | 11.21 | 18.53 | 12.97 |
| w/o grad. | 10.87 (3.0%↓) | 18.51 (0.1%↓) | 12.81 (1.2%↓) |
| w/o neigh. info | 8.44 (24.7%↓) | 13.06 (29.5%↓) | 11.25 (13.3%↓) |
| w/o both | 10.71 (4.5%↓) | 18.42 (0.6%↓) | 12.57 (3.1%↓) |

by the back-propagation. Comparing LightGCN with LightGCN$_{\text{w/o grad.}}$, we notice that the incorporation of gradient updates brought by MP is relatively incremental (i.e., ∼2%). However, to facilitate these additional gradient updates for slightly better performance, LightGCN is required to conduct MP at each batch, which brings tremendous additional overheads.

### 3.2 Message Passing in CF Helps Low-degree Users More Compared with High-degrees

Both empirical and theoretical evidence have demonstrated that GNNs usually perform satisfactorily on high-degree nodes with rich neighbor information but not as well on low-degree nodes [54, 23]. While designing graph-based model architectures for CF, most existing methods directly borrow this line of observations [60, 20] and assume that the contribution of message passing for CF is similar to that for graph learning tasks in general. However, whether or not these observations still transfer to message passing in CF remains questionable, as there exist architectural and philosophical gaps between message passing for CF and its counterparts for GNNs, as discussed in Section 2. To validate these hypotheses, we conduct experiments over representative methods (i.e., LightGCN and matrix factorization (MF) trained with BPR) and show their performance w.r.t. the node degree in Figure 1.

We observe that, overall both MF and LightGCN perform better on high-degree users than low-degree users. According to the upper two figures in Figure 1, MF behaves similarly to LightGCN, even without treating the input data as graphs, where the overall performance for high-degree user is stronger than that for low-degree users. However, the performance improvement of LightGCN from MF on low-degree users is larger than that for high-degree users (i.e., lower two figures in Figure 1). According to literature in general graph learning tasks [23, 36, 54], the performance improvement should be positively proportional to the node degree - the gain for high-degree users should be higher than that for low-degree users. This discrepancy indicates that it might not be appropriate to accredit contributions of message passing in CF directly through ideologies designed for classic graph learning tasks (e.g., node classification and link prediction). To bridge this gap, we connect supervision signals (i.e., BPR and DirectAU) commonly adopted by CF methods to Laplacian matrix learning. The formulation of BPR [45] and DirectAU [57] without the incorporation of graphs can be written as:

$$\mathcal{L}_{\text{BPR}} = -\sum_{(i,j)\in\mathcal{D}}\sum_{(i,k)\notin\mathcal{D}}\log\sigma(s_{ij}-s_{ik}) = -\sum_{(i,j)\in\mathcal{D}}\sum_{(i,k)\notin\mathcal{D}}\log\sigma(\mathbf{u}_i^\mathsf{T}\cdot\mathbf{i}_j - \mathbf{u}_i^\mathsf{T}\cdot\mathbf{i}_k),$$

$$\mathcal{L}_{\text{DirectAU}} = \sum_{(i,j)\in\mathcal{D}}||\mathbf{u}_i - \mathbf{i}_j||^2 + \sum_{u,u'\in\mathcal{U}}\log e^{-2||\mathbf{u}-\mathbf{u}'||^2} + \sum_{i,i'\in\mathcal{I}}\log e^{-2||\mathbf{i}-\mathbf{i}'||^2}, \quad (5)$$

where $\mathcal{D}$ refers to the set of observed interactions at the training phase and $\mathbf{i}'$ and $\mathbf{u}'$ refers to any random user/item. According to works on Laplacian matrix learning [82, 10, 38], learning node representations over graphs can be decoupled into Laplacian quadratic form, a weighted summation of two sub-goals:

$$\min_{\mathbf{Z}}\{||\mathbf{Z} - \mathbf{X}||^2 + \text{tr}(\mathbf{Z}^\mathsf{T}\mathbf{L}\mathbf{Z})\}, \quad (6)$$

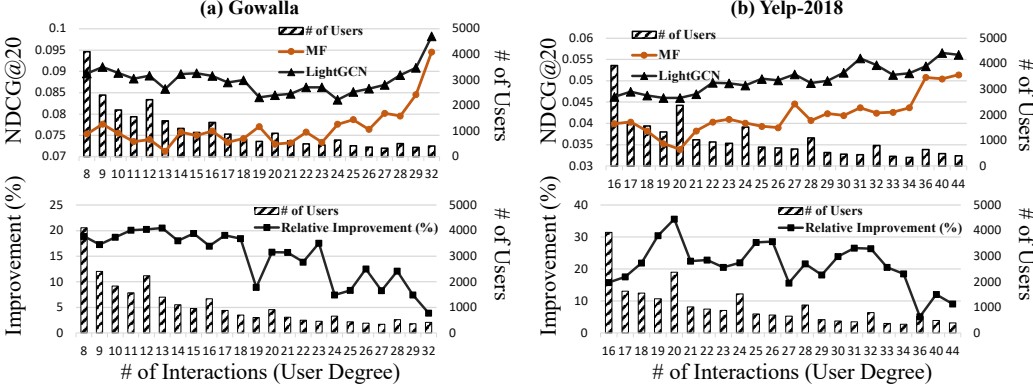

Figure 1: Performances of LightGCN and Matrix Factorization w.r.t. the user degree across datasets. The performance improvement brought by message passing decreases as the user degree goes up.

where $\mathbf{Z}$ refers to the node representation matrix after the message passing, $\mathbf{X}$ refers to the input feature matrix, and $\mathbf{L}$ refers to the Laplacian matrix. The first term regularizes the latent representation such that it does not diverge too much from the input feature; whereas the second term promotes the similarity between latent representations of adjacent nodes, which can be re-written as: $\mathrm{tr}(\mathbf{Z}^{\mathsf{T}} \cdot \mathbf{L} \cdot \mathbf{Z}) = \sum_{(i,j) \in \mathcal{D}} ||\mathbf{u}_i - \mathbf{i}_j||^2$ in CF bipartite graphs. [82] show that $K$ layers of linear message passing exactly optimizes the second term in Equation (6). Given this theoretical foundation, we derive the following theorem w.r.t. relations between BPR, DirectAU, and message passing in CF:

**Theorem 1.** *Assuming that* $||\mathbf{u}_i||^2 = ||\mathbf{i}_j||^2 = 1$ *for any* $u_i \in \mathcal{U}$ *and* $I_j \in \mathcal{I}$, *objectives of BPR and DirectAU are strictly upper-bounded by the objective of message passing (i.e.,* $\mathcal{L}_{BPR} \leq \sum_{(i,j) \in \mathcal{D}} ||\mathbf{u}_i - \mathbf{i}_j||^2$ *and* $\mathcal{L}_{DirectAU} \leq \sum_{(i,j) \in \mathcal{D}} ||\mathbf{u}_i - \mathbf{i}_j||^2$).

Proof of Theorem 1 can be found in Appendix A. According to Theorem 1, both BPR and DirectAU optimize the objective of message passing (i.e., $\sum_{(i,j) \in \mathcal{D}} ||\mathbf{u}_i - \mathbf{i}_j||^2$) with some additional regularization (i.e., dissimilarity between non-existing user/item pairs for BPR, and representation uniformity for DirectAU). Hence, directly optimizing these two objectives partially fulfills the effects brought by message passing during the back-propagation.

Combining this theory with the aforementioned empirical observations, we show that these two supervision signals could inadvertently conduct message passing in the backward step, even without explicitly treating interaction data as graphs. Since this inadvertent message passing happens during the back-propagation, its performance is positively correlated to the amount of training signals a user/item can get. In the case of CF, the amount of training signals for a user is directly proportional to the node degree. High-degree active users naturally benefit more from the inadvertent message passing from objective functions, because they acquire more training signals. Hence, when explicit message passing is applied to CF methods, the performance gain for high-degree users is less significant than that for low-degree users. Because the contribution of the message passing over high-degree nodes has been mostly fulfilled by the inadvertent message passing during the training.

To quantitatively prove this theory, we incrementally upsample low-degree training users and observe the performance improvement that TAG-CF could introduce at each upsampling rate. If our line of theory is correct, then we should expect less performance improvement on low-degree users for a larger upsampling rate. The results are shown in Appendix E with supporting evidence.

## 4  Test-time Aggregation for Collaborative Filtering

In Section 3, we demonstrate why message passing helps CF from two perspectives. Firstly, w.r.t. the formulation of LightGCN, we observe that the performance gain brought by neighbor information dominates that brought by additional gradients. Secondly, w.r.t. the improvement on user subgroups, we learn that message passing helps low-degree users more, compared with high-degree users.

In light of these two takeaways, we present **T**est-time **Ag**gregation for **C**ollaborative **F**iltering, namely **TAG-CF**, a test-time augmentation framework that only conducts message passing once at inference time and is effective at enhancing matrix factorization methods trained by different CF supervision signals. Given a set of well-trained user/item representations, TAG-CF simply aggregates neighboring item (user) representations for a given user (item) at test time. Despite its simplicity,

we show that our proposal can be used as a plug-and-play module and is effective at enhancing representations trained by different CF supervision signals.

The test-time aggregation is inspired by our first perspective that, within total performance gains brought by message passing, gains from additional neighbor representations during the forward pass dominate those brought by accompanying gradient updates to neighbors during the back-propagation. Applying message passing only at test time avoids repetitive training-time queries (i.e., once per node and epoch) of surrounding neighbors, which grow exponentially as the number of layers increases by the neighbor explosion phenomenon [16, 76, 75]. Specifically, given a set of well-trained user and item representations $\mathbf{U} \in \mathbb{R}^{|\mathcal{U}| \times d}$ and $\mathbf{I} \in \mathbb{R}^{|\mathcal{I}| \times d}$, TAG-CF augments representations for user $u_i$ and item $i_i$ as:

$$\mathbf{u}_i^* = \mathbf{u}_i + \sum_{i_j \in N(u_i)} |N(u_i)|^m |N(i_j)|^n \cdot \mathbf{i}_j, \mathbf{i}_i^* = \mathbf{i}_i + \sum_{u_j \in N(i_i)} |N(i_i)|^m |N(u_j)|^n \cdot \mathbf{u}_j, \quad (7)$$

where $m$ and $n$ are two hyper-parameters that control the normalization of message passing. With $m = n = -\frac{1}{2}$, Equation (7) becomes the exact formulation of one-layer LightGCN (i.e., Equation (2)). Empirically, we observe that the setup with $m = n = -\frac{1}{2}$ for TAG-CF does not always work for all datasets. This setup is directly migrated from message passing for homogeneous graphs [30], which might not be applicable for bipartite graphs where all neighbors are heterogeneous [7]. Unlike LightGCN which can fill this gap by adaptively tuning all representations during the training, TAG-CF cannot update any parameter since it is applied at test time, and hence requires tune-able normalization hyper-parameters.

Moreover, following our second perspective that message passing helps low-degree nodes more in CF, we further derive TAG-CF$^+$, which reduces the cost of TAG-CF by applying the one-time message passing only to low-degree nodes. Focusing on only low-degree nodes has two benefits: **(i)** it reduces the number of nodes that TAG-CF$^+$ needs to attend to, and **(ii)** message passing for low-degree nodes is naturally cheaper than for high-degree nodes given the surrounding neighborhoods are sparser (mitigating neighbor explosion). The degree threshold that determines which nodes to apply TAG-CF$^+$ is selected by the validation performance, with details in Appendix C.

TAG-CF can effectively enhance MF methods by conducting message passing only once at test time. TAG-CF effectively utilizes graphs while circumventing most of notorious computational overheads of message passing. It is extremely flexible, simple to implement, and enjoys the performance benefits of graph-based CF method while paying the lowest overall scalability.

## 5    Experiments

We conduct extensive experiments to demonstrate the effectiveness and efficiency of TAG-CF. We aim to answer the following research questions: **RQ (1)**: how effective is TAG-CF at improving MF methods without using graphs, **RQ (2)**: how much overheads does TAG-CF introduce, **RQ (3)**: can TAG-CF effectively enhance MF methods trained by different objectives, **RQ (4)**: how effective is TAG-CF$^+$ w.r.t. different degree cutoffs, and **RQ (5)**: do behaviors of TAG-CF align with our findings in Section 3?

### 5.1    Experimental Settings

**Datasets**. We conduct comprehensive experiments on five commonly used academic benchmark datasets, including `Amazon-book`, `Anime`, `Gowalla`, `Yelp2018`, and `MovieLens-1M`. Additionally, we also evaluate on a large-scale real-world industrial user-item recommendation dataset `Internal`. Descriptions of these datasets are provided in Appendix B.

**Baselines**. We compare TAG-CF with two branches of methods: (1) CF methods that do not explicitly utilize graphs, including vanilla matrix factorization (MF) methods trained by BPR and DirectAU [45, 57], Efficient Neural Matrix Factorization [3] (denoted as ENMF), and UltraGCN [39]. (2) Graph-based CF methods, including LightGCN [20] and NGCF [60]. Besides, we also compare with recent graph-based CF methods that extend LightGCN by adding additional self-supervised signals for better performance, including LightGCL [1], SimGCL [74], and SGL [65]. For the coherence of reading, we include comprehensive discussions about evaluation protocols across all methods, tuning for hyper-parameters, and other implementation details in Appendix C.

Table 2: Recommendation performance (i.e., NDCG@20 and Recall@20) of all models across users with different numbers of interactions. The lower percentile indicates the set of nodes whose degrees are ranked in the lower 30% population. **Bold** and underline indicate the best and second best model respectively. LightGCN and MF are trained with DirectAU [57].

| Method | NGCF | LightGCN | ENMF | +TAG-CF | Impr. (↑) | MF | +TAG-CF | Impr. (↑%) | UltraGCN | +TAG-CF | Impr. (↑%) |
|---|---|---|---|---|---|---|---|---|---|---|---|
| | | | | NDCG@20 – Low-degree Users (Lower Percentile) | | | | | | | |
| Amazon-Book | $5.32_{\pm0.08}$ | $\underline{8.09}_{\pm0.10}$ | $5.33_{\pm0.02}$ | $5.67_{\pm0.03}$ | 6.4% | $8.02_{\pm0.07}$ | $\mathbf{8.26}_{\pm0.06}$ | 3.0% | $5.61_{\pm0.19}$ | $6.04_{\pm0.21}$ | 7.7% |
| Anime | $20.13_{\pm0.18}$ | $27.78_{\pm0.21}$ | $22.23_{\pm0.19}$ | $22.58_{\pm0.15}$ | 1.6% | $23.95_{\pm0.07}$ | $27.15_{\pm0.04}$ | 13.4% | $\underline{28.14}_{\pm0.19}$ | $\mathbf{30.10}_{\pm0.21}$ | 7.0% |
| Gowalla | $8.46_{\pm0.06}$ | $\underline{10.08}_{\pm0.13}$ | $3.87_{\pm0.15}$ | $4.08_{\pm0.11}$ | 5.4% | $10.00_{\pm0.08}$ | $\mathbf{10.19}_{\pm0.04}$ | 1.9% | $8.21_{\pm0.09}$ | $8.63_{\pm0.11}$ | 5.1% |
| Yelp-2018 | $4.87_{\pm0.06}$ | $\underline{6.10}_{\pm0.09}$ | $3.11_{\pm0.07}$ | $3.26_{\pm0.04}$ | 4.8% | $6.08_{\pm0.08}$ | $\mathbf{6.18}_{\pm0.05}$ | 1.7% | $4.89_{\pm0.10}$ | $5.44_{\pm0.12}$ | 11.2% |
| MovieLens-1M | $22.13_{\pm0.26}$ | $25.95_{\pm0.28}$ | $18.34_{\pm0.19}$ | $22.53_{\pm0.21}$ | 22.8% | $20.98_{\pm0.12}$ | $\mathbf{29.20}_{\pm0.19}$ | 39.2% | $\underline{23.89}_{\pm0.19}$ | $28.37_{\pm0.21}$ | 18.8% |
| Internal | $5.91_{\pm0.07}$ | $\underline{8.12}_{\pm0.03}$ | OOM | - | - | $6.79_{\pm0.04}$ | $\mathbf{8.52}_{\pm0.06}$ | 25.5% | OOM | - | - |
| | | | | NDCG@20 – Overall | | | | | | | |
| Amazon-Book | $6.97_{\pm0.11}$ | $\underline{8.06}_{\pm0.11}$ | $6.13_{\pm0.13}$ | $6.54_{\pm0.09}$ | 6.7% | $8.01_{\pm0.03}$ | $\mathbf{8.13}_{\pm0.03}$ | 1.5% | $5.77_{\pm0.25}$ | $6.11_{\pm0.27}$ | 5.9% |
| Anime | $22.54_{\pm0.25}$ | $27.97_{\pm0.21}$ | $30.17_{\pm0.09}$ | $\underline{30.86}_{\pm0.12}$ | 2.3% | $24.01_{\pm0.06}$ | $27.25_{\pm0.03}$ | 9.8% | $30.30_{\pm0.11}$ | $\mathbf{30.89}_{\pm0.11}$ | 1.9% |
| Gowalla | $8.65_{\pm0.10}$ | $\mathbf{9.96}_{\pm0.11}$ | $5.23_{\pm0.04}$ | $5.29_{\pm0.05}$ | 1.1% | $9.77_{\pm0.08}$ | $\underline{9.88}_{\pm0.04}$ | 1.1% | $8.53_{\pm0.14}$ | $9.02_{\pm0.15}$ | 5.7% |
| Yelp-2018 | $5.54_{\pm0.06}$ | $\underline{6.33}_{\pm0.06}$ | $3.79_{\pm0.09}$ | $3.89_{\pm0.05}$ | 2.6% | $6.25_{\pm0.06}$ | $\mathbf{6.36}_{\pm0.03}$ | 1.8% | $5.01_{\pm0.11}$ | $5.53_{\pm0.11}$ | 10.4% |
| MovieLens-1M | $23.17_{\pm0.18}$ | $26.64_{\pm0.23}$ | $20.57_{\pm0.18}$ | $22.98_{\pm0.20}$ | 11.7% | $22.51_{\pm0.14}$ | $\mathbf{29.65}_{\pm0.17}$ | 31.7% | $26.50_{\pm0.15}$ | $\underline{29.68}_{\pm0.21}$ | 12.0% |
| Internal | $6.94_{\pm0.06}$ | $\underline{8.10}_{\pm0.06}$ | OOM | - | - | $7.04_{\pm0.02}$ | $\mathbf{8.54}_{\pm0.02}$ | 21.3% | OOM | - | - |
| | | | | Recall@20 – Low-degree Users (Lower Percentile) | | | | | | | |
| Amazon-Book | $10.71_{\pm0.14}$ | $\underline{13.18}_{\pm0.17}$ | $10.42_{\pm0.16}$ | $11.08_{\pm0.11}$ | 6.3% | $13.07_{\pm0.09}$ | $\mathbf{13.37}_{\pm0.10}$ | 2.3% | $7.92_{\pm0.15}$ | $8.31_{\pm0.10}$ | 4.9% |
| Anime | $25.74_{\pm0.35}$ | $32.74_{\pm0.21}$ | $\underline{37.14}_{\pm0.59}$ | $\mathbf{38.41}_{\pm0.53}$ | 3.4% | $29.08_{\pm0.09}$ | $31.94_{\pm0.05}$ | 9.8% | $33.96_{\pm0.28}$ | $36.49_{\pm0.28}$ | 7.4% |
| Gowalla | $17.53_{\pm0.32}$ | $\underline{19.14}_{\pm0.20}$ | $8.73_{\pm0.08}$ | $9.01_{\pm0.06}$ | 3.2% | $18.92_{\pm0.19}$ | $\mathbf{19.17}_{\pm0.13}$ | 1.3% | $15.57_{\pm0.18}$ | $16.01_{\pm0.15}$ | 2.8% |
| Yelp-2018 | $10.15_{\pm0.13}$ | $\underline{10.75}_{\pm0.14}$ | $7.17_{\pm0.06}$ | $7.54_{\pm0.12}$ | 5.2% | $10.63_{\pm0.13}$ | $\mathbf{10.98}_{\pm0.14}$ | 3.3% | $7.71_{\pm0.15}$ | $8.59_{\pm0.18}$ | 11.4% |
| MovieLens-1M | $22.71_{\pm0.16}$ | $25.80_{\pm0.22}$ | $19.58_{\pm0.14}$ | $24.11_{\pm0.16}$ | 23.1% | $23.64_{\pm0.12}$ | $\mathbf{28.10}_{\pm0.20}$ | 18.9% | $26.13_{\pm0.21}$ | $\mathbf{28.97}_{\pm0.23}$ | 10.9% |
| Internal | $10.54_{\pm0.09}$ | $\underline{13.81}_{\pm0.02}$ | OOM | - | - | $11.13_{\pm0.05}$ | $\mathbf{13.97}_{\pm0.06}$ | 25.5% | OOM | - | - |
| | | | | Recall@20 – Overall | | | | | | | |
| Amazon-Book | $10.30_{\pm0.21}$ | $\underline{12.76}_{\pm0.18}$ | $10.89_{\pm0.18}$ | $11.35_{\pm0.09}$ | 4.2% | $12.67_{\pm0.06}$ | $\mathbf{12.97}_{\pm0.06}$ | 2.4% | $8.01_{\pm0.25}$ | $8.53_{\pm0.27}$ | 6.5% |
| Anime | $28.12_{\pm0.22}$ | $32.82_{\pm0.21}$ | $34.10_{\pm0.25}$ | $34.48_{\pm0.23}$ | 1.1% | $29.15_{\pm0.09}$ | $31.95_{\pm0.05}$ | 6.9% | $\underline{35.87}_{\pm0.39}$ | $\mathbf{37.01}_{\pm0.39}$ | 3.2% |
| Gowalla | $17.93_{\pm0.06}$ | $\mathbf{18.65}_{\pm0.14}$ | $9.68_{\pm0.06}$ | $9.74_{\pm0.09}$ | 0.6% | $18.30_{\pm0.17}$ | $\underline{18.53}_{\pm0.11}$ | 1.3% | $15.93_{\pm0.21}$ | $16.36_{\pm0.22}$ | 2.7% |
| Yelp-2018 | $10.02_{\pm0.06}$ | $\underline{10.98}_{\pm0.10}$ | $6.89_{\pm0.09}$ | $7.05_{\pm0.03}$ | 2.3% | $10.81_{\pm0.10}$ | $\mathbf{11.21}_{\pm0.09}$ | 3.7% | $8.41_{\pm0.19}$ | $9.89_{\pm0.20}$ | 17.6% |
| MovieLens-1M | $23.93_{\pm0.14}$ | $26.30_{\pm0.20}$ | $21.31_{\pm0.19}$ | $23.88_{\pm0.25}$ | 12.1% | $26.30_{\pm0.14}$ | $\underline{28.40}_{\pm0.15}$ | 8.0% | $27.14_{\pm0.19}$ | $\mathbf{29.78}_{\pm0.23}$ | 9.7% |
| Internal | $6.91_{\pm0.04}$ | $\underline{13.89}_{\pm0.06}$ | OOM | - | - | $11.83_{\pm0.02}$ | $\mathbf{14.41}_{\pm0.08}$ | 21.8% | OOM | - | - |

Table 3: Running time ($1 \times 10^3$ seconds) for MF methods and TAG-CF. Time % is the percentage of running time TAG-CF takes w.r.t. the time for corresponding MF methods. Speed↑ refers to the ratio of running times between training-time aggregation (i.e., LightGCN) and TAG-CF. All training steps are timed and terminated by an early stopping strategy (see Appendix C).

| Method | Sparsity | ENMF | +TAG-CF | Time % | UltraGCN | +TAG-CF | Time % | LightGCN | MF | +TAG-CF | Time % | Speed↑ |
|---|---|---|---|---|---|---|---|---|---|---|---|---|
| Anime | 99.13% | 12.31 | +0.04 | 0.3% | 93.31 | +0.04 | 0.1% | 138.85 | 34.12 | +0.04 | 0.3% | 4.06× |
| Yelp-2018 | 99.87% | 2.15 | +0.02 | 0.9% | 5.02 | +0.02 | 0.4% | 5.81 | 3.17 | +0.02 | 0.6% | 1.83× |
| Gowalla | 99.91% | 4.56 | +0.02 | 0.4% | 12.55 | +0.02 | 0.2% | 13.27 | 7.74 | +0.02 | 0.3% | 1.72× |
| Amazon-Book | 99.94% | 11.54 | +0.03 | 0.3% | 39.25 | +0.03 | 0.1% | 46.62 | 29.21 | +0.03 | 0.1% | 1.59× |
| Internal | 99.99% | OOM | - | - | OOM | - | - | 47.32 | 32.62 | + 0.09 | 0.3% | 1.44× |

## 5.2 Performance Improvement to Matrix Factorization Methods

For **RQ (1)**, Table 2 shows the performances of MF methods (MF and ENMF) as well as that of the performances of them with TAG-CF applied on their learned representations. We observe that TAG-CF unanimously improves the recommendation performance for both of them. Specifically, across all datasets, TAG-CF on average improves the low-degree NDCG@20 by 4.6% and 9.1% and overall NDCG by 3.2% and 7.1% for ENMF and MF, respectively. We also observe a similar performance improvement for Recall1@20, where TAG-CF on average improves the low-degree Recall@20 by 4.5% and 8.4% and overall Recall@20 by 2.1% and 7.2% for ENMF and MF, respectively. Furthermore, we notice that TAG-CF can improve the performance of UltraGCN, a method that utilizes the graph knowledge as additional supervision signals. This phenomenon demonstrates the superior effectiveness of TAG-CF, indicating that our proposed test-time aggregation can further enhance graph-enhanced MF methods.

By comparing the performance gains brought by TAG-CF on low-degree users with that on all users, we notice that gains for low-degree users are usually higher. Hence, message passing in CF helps low-degree users more than for high-degree users, which echos with our observations in Section 3.1. To answer **RQ (5)**, the behavior of TAG-CF aligns with our second perspective in Section 3.2 that the supervision signal inadvertently conducts message passing. Consequently, the room for improvement on high-degree users could be limited, as part of the contributions from message passing has already been claimed by the supervision signal.

Table 4: The running time and performance of graph-based CF methods that extend LightGCN.

| Method | SGL | SimGCL | LightGCL | TAG-CF |
|---|---|---|---|---|
| | NDCG@20 – OVERALL | | | |
| Anime | $27.02_{\pm0.05}$ | $30.48_{\pm0.12}$ | $28.34_{\pm0.16}$ | $27.25_{\pm0.03}$ |
| Yelp | $5.67_{\pm0.04}$ | $5.99_{\pm0.09}$ | $4.93_{\pm0.06}$ | $6.36_{\pm0.03}$ |
| Gowalla | $9.67_{\pm0.17}$ | $10.32_{\pm0.06}$ | $8.99_{\pm0.13}$ | $9.88_{\pm0.04}$ |
| Book | $6.69_{\pm0.02}$ | $7.02_{\pm0.05}$ | $5.83_{\pm0.08}$ | $8.13_{\pm0.04}$ |
| Avg. Rank | 3.2 | _1.7_ | 3.5 | **1.2** |
| | RECALL@20 – OVERALL | | | |
| Anime | $31.29_{\pm0.09}$ | $34.93_{\pm0.14}$ | $33.64_{\pm0.22}$ | $31.95_{\pm0.05}$ |
| Yelp | $10.01_{\pm0.08}$ | $10.56_{\pm0.13}$ | $8.83_{\pm0.04}$ | $11.21_{\pm0.09}$ |
| Gowalla | $18.18_{\pm0.24}$ | $19.22_{\pm0.09}$ | $16.99_{\pm0.10}$ | $18.53_{\pm0.09}$ |
| Book | $11.15_{\pm0.04}$ | $11.51_{\pm0.09}$ | $10.06_{\pm0.05}$ | $12.97_{\pm0.06}$ |
| Avg. Rank | 3.2 | **1.5** | 3.5 | _1.7_ |
| | RUNNING TIME ($1 \times 10^3$ SECOND) | | | |
| Anime | 69.48 | 87.77 | 97.31 | 34.15 |
| Yelp | 3.94 | 9.72 | 4.30 | 3.19 |
| Gowalla | 9.32 | 29.11 | 11.10 | 7.76 |
| Book | 63.21 | 71.39 | 38.87 | 29.24 |
| Avg. Rank | _2.2_ | 3.8 | 3.0 | **1.0** |
| Total Rank | 3.6 | _2.8_ | 3.9 | **1.9** |

Table 5: Performance of TAG-CF when applied to models trained with BPR loss.

| Method | LightGCN | MF | +TAG-CF | Impr. (↑%) |
|---|---|---|---|---|
| | NDCG@20 – LOW-DEGREE USERS (LOWER PERCENTILE) | | | |
| Anime | $30.02_{\pm0.07}$ | $29.36_{\pm0.23}$ | $30.56_{\pm0.27}$ | 4.1% |
| Yelp | $4.34_{\pm0.07}$ | $3.63_{\pm0.15}$ | $3.81_{\pm0.18}$ | 5.0% |
| Gowalla | $8.22_{\pm0.03}$ | $7.56_{\pm0.14}$ | $7.88_{\pm0.15}$ | 4.2% |
| Book | $5.19_{\pm0.14}$ | $4.19_{\pm0.14}$ | $4.68_{\pm0.14}$ | 11.7% |
| | NDCG@20 – OVERALL | | | |
| Anime | $30.14_{\pm0.07}$ | $29.51_{\pm0.21}$ | $30.23_{\pm0.26}$ | 2.4% |
| Yelp | $4.87_{\pm0.06}$ | $3.96_{\pm0.14}$ | $4.26_{\pm0.17}$ | 7.6% |
| Gowalla | $8.32_{\pm0.03}$ | $7.51_{\pm0.12}$ | $7.99_{\pm0.14}$ | 6.4% |
| Book | $5.07_{\pm0.15}$ | $4.15_{\pm0.13}$ | $4.32_{\pm0.13}$ | 4.1% |
| | RECALL@20 – LOW-DEGREE USERS (LOWER PERCENTILE) | | | |
| Anime | $34.23_{\pm0.08}$ | $34.81_{\pm0.32}$ | $35.42_{\pm0.35}$ | 1.8% |
| Yelp | $8.19_{\pm0.20}$ | $6.93_{\pm0.26}$ | $7.25_{\pm0.19}$ | 4.6% |
| Gowalla | $16.17_{\pm0.12}$ | $14.86_{\pm0.23}$ | $15.33_{\pm0.24}$ | 3.2% |
| Book | $8.81_{\pm0.26}$ | $7.45_{\pm0.22}$ | $8.05_{\pm0.15}$ | 8.1% |
| | RECALL@20 – OVERALL | | | |
| Anime | $34.21_{\pm0.08}$ | $34.84_{\pm0.30}$ | $35.23_{\pm0.34}$ | 1.1% |
| Yelp | $8.33_{\pm0.30}$ | $7.27_{\pm0.27}$ | $7.62_{\pm0.22}$ | 4.8% |
| Gowalla | $15.69_{\pm0.07}$ | $14.47_{\pm0.23}$ | $14.92_{\pm0.25}$ | 3.1% |
| Book | $8.65_{\pm0.24}$ | $7.35_{\pm0.22}$ | $7.64_{\pm0.20}$ | 3.9% |

## 5.3 Performance Comparison Among Graph-based Methods

Comparing TAG-CF with LightGCN in Table 2, we can notice that TAG-CF mostly performs on par with and sometimes even outperforms LightGCN, without incorporating message passing during the training and only conducting test-time aggregation. This phenomenon indicates that conducting neighbor aggregation at the testing time can recover most of the contributions of training-time message passing. To answer **RQ (5)**, TAG-CF aligns with our first perspective in Section 3.1 that the performance gain from beneficial neighbor information dominates their accompanying gradients.

We further compare TAG-CF with state-of-the-art graph-based CF methods, with their performance and efficiency shown in Table 4. Among these performant baselines, TAG-CF exhibits competitive performance, with an average rank of 1.2 on NDCG and 1.7 on Recall. Though not always the model that delivers the best performance, TAG-CF can deliver comparably promising results and introduces little computational overheads (i.e., ranked 1.0 for running time). Considering efficiency as one factor, TAG-CF achieves the best performance across all baselines with an average rank of 1.9.

While performing on par with graph-based CF methods that aggregate neighbor contents at the training time, TAG-CF enjoys the performance benefits of message passing while paying the lowest overall scalability. To answer **RQ (2)**, according to Table 3, across all datasets, TAG-CF only introduces an average additional computational overhead of $0.05 \times 10^3$ seconds, which is less than 0.5% of the total training time for matrix factorization methods. Comparing the running time of LightGCN with that of TAG-CF, we can observe that the latter can significantly improve the computational time, and the speedup is proportional to the sparsity of the dataset.

## 5.4 Effectiveness for Different Training Signals

To answer **RQ (3)**, besides DirectAU, we also conduct experiments on BPR loss, as shown in Table 5. When applied to BPR, TAG-CF still consistently improves the performance by large margins (i.e., 6.3% and 5.1% average improvement on low-degree and overall NDCG respectively, and 4.4% and 3.2% on low-degree and overall Recall respectively). We notice that TAG-CF sometimes does not perform as competitively as LightGCN when both are trained with BPR. We check norms of learned representations from MF with BPR and discover that they have high variance since BPR does not explicitly enforce any regularization. This might not favor TAG-CF as a test-time augmentation method due to its simple design, which cannot adapt representations with high variance.

## 5.5 Performance w.r.t. User Degree

To answer **RQ (4)**, we apply TAG-CF$^+$ to four public datasets and the performance and the efficiency improvement are demonstrated in Figure 2. Overall, the running time improvement brought by TAG-CF$^+$ exponentially increases as the degree decreases, since low-degree users have sparse neighborhoods and there is hence less information for TAG-CF$^+$ to aggregation. When the degree cutoff is low (i.e., less than 100), the effectiveness of TAG-CF$^+$ proportional increases as the degree cutoff increases.

When setting the cutoff to a user degree of around 100, on `Amazon-Book`, `Gowalla`, and `Yelp-2018`, TAG-CF$^+$ can further improve TAG-CF by 125%, 17%, and 11%, respectively, with efficiency improvement of 7%, 4%, and 8%. In these cases, TAG-CF$^+$ not only significantly improves the performance but also effectively reduces computational overheads.

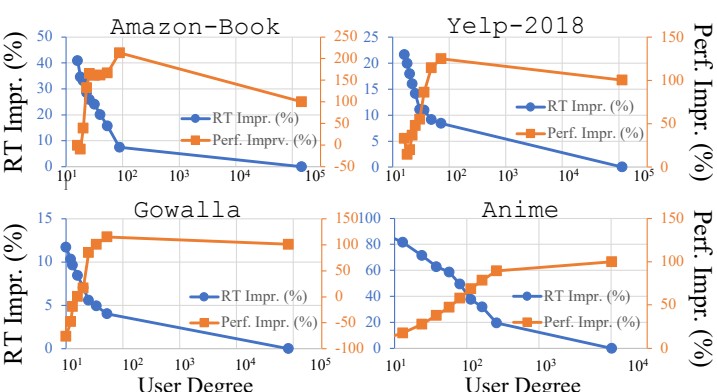

However, on these three datasets, after the cutoff bypasses a degree of 100, the performance improvement eventually decreases to the performance of TAG-CF (i.e., 100%), indicating that test-time aggregation jeopardizes the performance on high-degree nodes. On `Anime`, though no downgrade on high-degree users, the performance improvement of TAG-CF$^+$ to TAG-CF is incremental. These phenomenons not only demonstrate the effectiveness and efficiency of TAG-CF$^+$, but also verify our findings in Section 3.2 that message passing in CF helps low-degree users more than high-degree users.

Figure 2: The performance and efficiency improvement of TAG-CF$^+$ w.r.t. different cutoffs. TAG-CF$^+$ further improves TAG-CF with less computational overheads. 100% is the original performance/efficiency of vanilla TAG-CF.

## 6 Conclusion

In this study, we investigate how message passing improves collaborative filtering. Through a series of ablations, we demonstrate that the performance gain from neighbor contents dominates that from accompanying gradients brought by message passing in CF. Moreover, for the first time, we show that message passing in CF improves low-degree users more than high-degree users. We theoretically demonstrate that CF supervision signals inadvertently conduct message passing in the backward step, even without treating the data as a graph. In light of these novel takeaways, we propose TAG-CF, a test-time aggregation framework effective at enhancing representations trained by different CF supervision signals. Evaluated on five datasets, TAG-CF performs at par with SoTA methods with only a fraction of computational overhead (i.e., less than 1.0% of the total training time).

## 7 Limitation and Broader Impact

One limitation of our proposal could be the utilization of graphs in large-scale machine learning pipeline. TAG-CF conducts a single-time aggregation of neighbors, which could be equivalently achieved by existing technologies such as SQL, BigQuery, etc. Furthermore, we observe no ethical concern entailed by our proposal, but we note that both ethical or unethical applications based on collaborative filtering may benefit from the effectiveness of our work. Care should be taken to ensure socially positive and beneficial results of machine learning algorithms.

## 8 Acknowledgments

This work was mostly conducted during the internship of Clark, William, and Zhichun at Snap Inc. We would like to thank Xin Chen and his colleagues from Snap Inc. for their help on pre-processing the internal dataset. This work was partially supported by the NSF under grants IIS-2321504, IIS-2334193, IIS-2203262, IIS-2217239, CNS-2426514, CNS-2203261, and CMMI-2146076. Any opinions, findings, and conclusions or recommendations expressed in this material are those of the authors and do not necessarily reflect the views of the sponsors.

We sincerely appreciate constructive feedback from all reviewers during the paper review phase.

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

# A  Proof of Theorem 1

Here we re-state Theorem 1 before diving into its proof:

**Theorem 1.** Given that a K-layer GCN minimizes $\sum_{(i,j)\in\mathcal{D}} ||\mathbf{u}_i - \mathbf{i}_j||^2$, During the training of MF methods, assuming that $||\mathbf{u}_i||^2 = ||\mathbf{i}_j||^2 = 1$ for any $u_i \in \mathcal{U}$ and $I_j \in \mathcal{I}$, objectives of BPR and DirectAU are strictly upper-bounded by the objective of message passing (i.e., $\mathcal{L}_{\text{BPR}} \leq \sum_{(i,j)\in\mathcal{D}} ||\mathbf{u}_i - \mathbf{i}_j||^2$ and $\mathcal{L}_{\text{DirectAU}} \leq \sum_{(i,j)\in\mathcal{D}} ||\mathbf{u}_i - \mathbf{i}_j||^2$).

One preliminary theoretical foundation for Theorem 1 to hold is that a K-layer graph convolution network (GCN) exactly optimizes the second term in Equation (6), which has been proved by [82]. For ease of reading, we re-phrase it again as the following:

**Theorem 2.** *The message passing for GCN optimizes the following graph regularization term:* $\mathcal{O} = \min_{\mathbf{Z}}\{tr(\mathbf{Z}^\intercal \mathbf{L}\mathbf{Z}))\}$.

*Proof.* Set derivative of $tr(\mathbf{Z}^\intercal \mathbf{L}\mathbf{Z})$ with respect to $\mathbf{Z}$ to zero:

$$\frac{\partial tr(\mathbf{Z}^\intercal \mathbf{L}\mathbf{Z})}{\partial \mathbf{Z}} = 0 \rightarrow \mathbf{L}\mathbf{Z} = 0 \rightarrow \mathbf{Z} = A\mathbf{Z}. \tag{8}$$

With K$\to\infty$:

$$\mathbf{Z}^{(K)} = \mathbf{A}\mathbf{Z}^{(K-1)} \tag{9}$$

which indicates:

$$\mathbf{Z}^{(K)} = \mathbf{A}\mathbf{Z}^{(K-1)} = \mathbf{A}^2\mathbf{Z}^{(K-2)} = \cdots = \mathbf{A}^K\mathbf{Z}^{(0)} = \mathbf{A}^K\mathbf{X}\mathbf{W}. \tag{10}$$

$\square$

According to this theoretical foundation, it is straightforward that Theorem 2 is also applicable for the message passing of LightGCN in the setting of CF if we let $\mathbf{A} = \{0,1\}^{(|\mathcal{U}|+|\mathcal{I}|)\times(|\mathcal{U}|+|\mathcal{I}|)}$, $\mathbf{X} = \mathbf{I}_{|\mathcal{U}|+|\mathcal{I}|}$, and $\mathbf{W} = (\mathbf{U}||\mathbf{I})$, where $||$ refers to the concatenation operation. With this preliminary, the proof to Theorem 1 starts as:

*Proof.* DirectAU optimizes:

$$\mathcal{L}_{\text{DirectAU}} = \sum_{(i,j)\in\mathcal{D}} ||\mathbf{u}_i - \mathbf{i}_j||^2 \tag{11}$$

$$+ \sum_{u,u'\in\mathcal{U}} \log e^{-2||\mathbf{u}-\mathbf{u}'||^2} + \sum_{i,i'\in\mathcal{I}} \log e^{-2||\mathbf{i}-\mathbf{i}'||^2}. \tag{12}$$

Since $\sum_{u,u'\in\mathcal{U}} \log e^{-2||\mathbf{u}-\mathbf{u}'||^2} <= 0$ and $\sum_{i,i'\in\mathcal{I}} \log e^{-2||\mathbf{i}-\mathbf{i}'||^2} <= 0$, we directly have $\mathcal{L}_{\text{DirectAU}} \leq \sum_{(i,j)\in\mathcal{D}} ||\mathbf{u}_i - \mathbf{i}_j||^2$.

BPR optimizes:

$$\mathcal{L}_{\text{BPR}} = -\sum_{(i,j)\in\mathcal{D}} \sum_{(i,k)\notin\mathcal{D}} \log \sigma(s_{ij} - s_{ik}) = \tag{13}$$

$$-\sum_{(i,j)\in\mathcal{D}} \sum_{(i,k)\notin\mathcal{D}} \log \sigma(\mathbf{u}_i^\intercal \cdot \mathbf{i}_j - \mathbf{u}_i^\intercal \cdot \mathbf{i}_k) \tag{14}$$

$$= \sum_{(i,j)\in\mathcal{D}} \sum_{(i,k)\notin\mathcal{D}} -\log \left(\frac{e^{\mathbf{u}_i^\intercal \cdot \mathbf{i}_j}}{e^{\mathbf{u}_i^\intercal \cdot \mathbf{i}_j} + e^{\mathbf{u}_i^\intercal \cdot \mathbf{i}_k}}\right) \tag{15}$$

$$= \sum_{(i,j)\in\mathcal{D}} \sum_{(i,k)\notin\mathcal{D}} -\mathbf{u}_i^\intercal \cdot \mathbf{i}_j + \log \left(e^{\mathbf{u}_i^\intercal \cdot \mathbf{i}_j} + e^{\mathbf{u}_i^\intercal \cdot \mathbf{i}_k}\right) \tag{16}$$

Since $||\mathbf{u}_i||^2 = ||\mathbf{i}_j||^2 = 1$ for any $u_i \in \mathcal{U}$ and $I_j \in \mathcal{I}$, $||\mathbf{u}_i - \mathbf{i}_j|| = \sqrt{1 - 2\mathbf{u}_i^\intercal \cdot \mathbf{i}_j + 1} \rightarrow -\mathbf{u}_i^\intercal \cdot \mathbf{i}_j = \frac{1}{2}||\mathbf{u}_i - \mathbf{i}_j||^2 - 1$. So Equation (16) can be written as:

$$\mathcal{L}_{\text{BPR}} = \frac{1}{2}||\mathbf{u}_i - \mathbf{i}_j||^2 - 1 + \log \left(e^{\mathbf{u}_i^\intercal \cdot \mathbf{i}_j} + e^{\mathbf{u}_i^\intercal \cdot \mathbf{i}_k}\right). \tag{17}$$

The maximum possible value of $e^{\mathbf{u}_i^\intercal \cdot \mathbf{i}_j} + e^{\mathbf{u}_i^\intercal \cdot \mathbf{i}_k}$ is $2e$, which is less than 10. Hence $\log \left( e^{\mathbf{u}_i^\intercal \cdot \mathbf{i}_j} + e^{\mathbf{u}_i^\intercal \cdot \mathbf{i}_k} \right) < 1$, which leads to the second part of Theorem 1: $\mathcal{L}_{\text{BPR}} \leq \sum_{(i,j) \in \mathcal{D}} ||\mathbf{u}_i - \mathbf{i}_j||^2$. $\qquad\square$

# B  Dataset Description and Statistics

We conduct experiments on five commonly used benchmark datasets, that have been broadly utilized by the recommender system community, including `Amazon-book` [40], `Anime` [29], `Gowalla` [5], `Yelp2018` [71], and `MovieLens-1M` [19]. Additionally, we also evaluate our method on a large-scale industrial user-content recommendation dataset - `Internal`, with statistics shown in Table 6.

Table 6: Statistics of datasets explored in this work. Due to privacy constrains, we only report approximated values for `Internal` dataset.

| Dataset | # Users | # Items | # Interactions | Sparsity |
|---|---|---|---|---|
| `Amazon-book` | 52,643 | 40,981 | 2,984,108 | 99.94% |
| `Anime` | 73,515 | 12,295 | 7,813,727 | 99.13% |
| `Gowalla` | 29,858 | 40,981 | 1,027,370 | 99.91% |
| `Yelp-2018` | 31,668 | 38,048 | 1,561,406 | 99.87% |
| `MovieLens-1M` | 6,040 | 3,629 | 836,478 | 96.18% |
| `Internal` | ~0.5M | ~0.2M | ~7M | 99.99% |

# C  Additional Experimental Settings

## C.1  Evaluation Protocol

We evaluate all models using metrics adopted in previous works, including NDCG@20 and Recall@20 [20]. For the dataset split, we conduct the group-by-user splits and randomly select 80%, 10%, and 10% of observed interactions as training, validation, and testing sets respectively. We adopt an early stopping strategy, where the training will be terminated if the validation NDCG@20 stops increasing for 3 continuous epochs. We use models with the best validation performance to report the performance. Besides, the evaluation metrics are computed by the all-ranking protocol, where all items are listed as candidates [45]. We explore this strategy since we want to evaluate the representation quality of all users. All experiments are conducted 10 times with different seeds, and we report both means and standard deviations across independent runs.

## C.2  Hyper-parameter Tuning

We only conduct 25 searches per model for all methods to ensure the comparison fairness, so that our experiments are not biased to methods with sophisticated hyper-parameter search spaces. Furthermore, we set the embedding dimensions for all models to 64 (i.e., $d = 64$) to ensure a fair comparison, since a larger dimension usually leads to better performance in CF methods. For TAG-CF, we only tune $m$ and $n$ in Equation (7) during test time from the list of [-2, -1.5, -1, -0.5, 0]. Besides, we train all models using Adam optimizer. TAG-CF's sensitivty to $m$ and $n$ is visually plotted in Figure 4. We can observe that m and n are important for the success of TAG-CF. Fortunately, across datasets, the optimal selection of m and n is pretty similar (e.g., m=n=-0.5 or m=n=0). The other solution to automatically tune m and n could be initialzing m and n to -0.5 (i.e., the value that generally works well across datasets) and conducting gradient descent on them using the training loss. But in this work we observe that manually tuning them on a small set of candidates can already deliver promising results.

## C.3  Implementation Detail

We conduct most of the baseline experiments with RecBole [81]. Besides, we use Google Cloud Platform with 12 CPU cores, 64GB RAM, and a single V100 GPU with 16GB VRAM to run all experiments.

Table 7: Improvement of TAG-CF$^+$ to TAG-CF. Degree cutoffs are selected according to Figure 3.

| Metric | Yelp-2018 | Gowalla | Amazon-book | Anime |
|---|---|---|---|---|
| BPR | | | | |
| NDCG@20 | 27.1% | 10.3% | 122.4% | 0% |
| Recall@20 | 31.4% | 14.2% | 119.2% | 0% |
| Running Time | 8% | 4% | 9% | 0% |
| DIRECTAU | | | | |
| NDCG@20 | 34.1% | 22.5% | 98.3% | 0% |
| Recall@20 | 29.2% | 30.1% | 104.1% | 0% |
| Running Time | 8% | 4% | 9% | 0% |

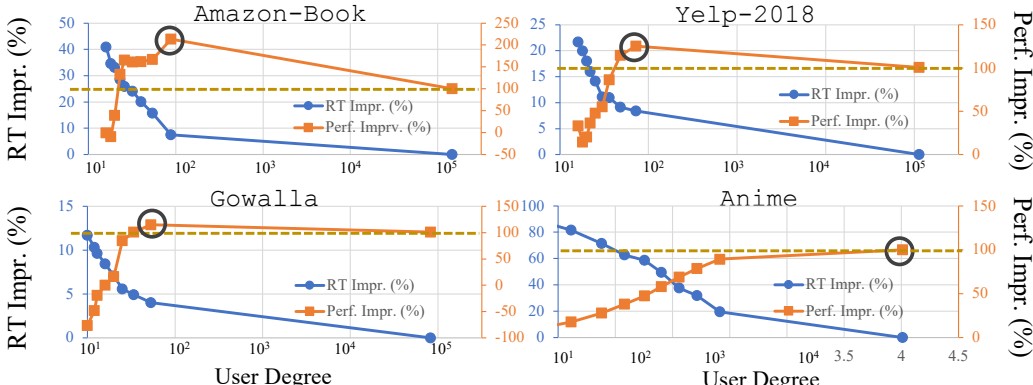

Figure 3: Improvement of TAG-CF$^+$ w.r.t. different cutoffs. Yellow dashed lines indicate TAG-CF, and black circles refer to the optimal degree cutoff that TAG-CF$^+$ selects.

# D   Degree Cutoff Selection for TAG-CF+

We first sort all users according to their degree and split the sorted list into 10 user buckets[3], where each bucket contains non-overlapped users with similar degrees. Starting from the bucket with the lowest user degree, TAG-CF$^+$ keeps applying test-time-aggregation demonstrated in Equation (7) to all buckets until the validation performance starts to decrease or the performance improvement is less than 2% compared with TAG-CF. The degree cutoffs circled in Figure 3 are the ones selected by this strategy and most of them correspond to the most performant configuration, shown in Table 7.

# E   Analysis of TAG-CF through Up-sampling

In Section 3, we connect CF objective functions to message passing and show that they inadvertently conduct message passing during the back-propagation. Since this inadvertent message passing happens during the back-propagation, its performance is positively correlated to the amount of training signals a user/item can get.

Table 8: Improvement (NDCG@20) brought by TAG-CF at different degree cutoffs and upsampling rates on ML-1M.

| Up-sampling Degree | Up-sample Rate: 100% | | | Up-sample Rate: 300% | | |
|---|---|---|---|---|---|---|
| | MF | + TAG-CF | Impr. (↑%) | MF | + TAG-CF | Impr. (↑%) |
| 40 | 20.62 | 28.87 | 38.8% | 19.30 | 25.01 | 30.3% |
| 80 | 20.10 | 27.43 | 35.9% | 18.40 | 23.30 | 26.8% |
| 160 | 19.39 | 26.63 | 36.6% | 17.93 | 23.37 | 29.8% |

In the case of CF, the amount of training signals for a user is directly proportional to the node degree of this user. High-degree active users naturally benefit more from the inadvertent message passing from objective functions like BPR and DirectAU, because they acquire more training signals from the objective function. Hence, when explicit message passing is applied to CF methods, the performance gain for high-degree users is less significant than that for

---

[3]The number of buckets can be set to arbitrary numbers for finer adjustments. In this study, we pick 10 as a proof of concept.

Table 9: Recommendation performance (i.e., NDCG@10 and Recall@10) of all models across users with different numbers of interactions. Setting explored in this table is the same as what Table 2 has.

| Method | NGCF | LightGCN | ENMF | +TAG-CF | Impr. (↑) | MF | +TAG-CF | Impr. (↑%) | UltraGCN | +TAG-CF | Impr. (↑%) |
|---|---|---|---|---|---|---|---|---|---|---|---|
| NDCG@10 – LOW-DEGREE USERS (LOWER PERCENTILE) | | | | | | | | | | | |
| Amazon-Book | $3.33_{\pm0.09}$ | $5.05_{\pm0.11}$ | $3.32_{\pm0.02}$ | $3.54_{\pm0.03}$ | 6.8% | $5.01_{\pm0.08}$ | $5.19_{\pm0.06}$ | 3.7% | $3.49_{\pm0.17}$ | $3.80_{\pm0.24}$ | 8.7% |
| Anime | $7.10_{\pm0.18}$ | $9.77_{\pm0.17}$ | $7.88_{\pm0.21}$ | $8.03_{\pm0.12}$ | 1.9% | $8.48_{\pm0.07}$ | $9.68_{\pm0.03}$ | 14.2% | $9.98_{\pm0.16}$ | $10.69_{\pm0.24}$ | 7.0% |
| Gowalla | $5.54_{\pm0.07}$ | $6.59_{\pm0.12}$ | $2.54_{\pm0.17}$ | $2.68_{\pm0.09}$ | 5.4% | $6.54_{\pm0.07}$ | $6.73_{\pm0.04}$ | 2.9% | $5.38_{\pm0.10}$ | $5.70_{\pm0.11}$ | 6.0% |
| Yelp-2018 | $2.80_{\pm0.07}$ | $3.51_{\pm0.10}$ | $1.80_{\pm0.08}$ | $1.89_{\pm0.04}$ | 5.2% | $3.52_{\pm0.09}$ | $3.57_{\pm0.05}$ | 1.6% | $2.82_{\pm0.10}$ | $3.16_{\pm0.14}$ | 12.1% |
| MovieLens-1M | $15.14_{\pm0.30}$ | $17.81_{\pm0.32}$ | $12.55_{\pm0.19}$ | $15.55_{\pm0.22}$ | 23.9% | $14.44_{\pm0.11}$ | $20.11_{\pm0.17}$ | 39.2% | $16.39_{\pm0.17}$ | $19.54_{\pm0.23}$ | 19.2% |
| NDCG@10 – OVERALL | | | | | | | | | | | |
| Amazon-Book | $4.35_{\pm0.13}$ | $5.03_{\pm0.10}$ | $3.81_{\pm0.11}$ | $4.09_{\pm0.08}$ | 7.4% | $4.98_{\pm0.03}$ | $5.08_{\pm0.03}$ | 1.9% | $3.60_{\pm0.25}$ | $3.83_{\pm0.23}$ | 6.6% |
| Anime | $7.99_{\pm0.27}$ | $9.92_{\pm0.24}$ | $10.63_{\pm0.10}$ | $10.97_{\pm0.11}$ | 3.1% | $8.52_{\pm0.05}$ | $9.73_{\pm0.03}$ | 14.2% | $10.66_{\pm0.10}$ | $10.99_{\pm0.09}$ | 3.1% |
| Gowalla | $5.65_{\pm0.11}$ | $6.54_{\pm0.11}$ | $3.42_{\pm0.05}$ | $3.49_{\pm0.05}$ | 2.1% | $6.41_{\pm0.07}$ | $6.53_{\pm0.04}$ | 1.9% | $5.59_{\pm0.14}$ | $5.93_{\pm0.14}$ | 6.1% |
| Yelp-2018 | $3.19_{\pm0.05}$ | $3.64_{\pm0.05}$ | $2.18_{\pm0.11}$ | $2.26_{\pm0.05}$ | 3.6% | $3.59_{\pm0.07}$ | $3.67_{\pm0.03}$ | 2.3% | $2.88_{\pm0.11}$ | $3.21_{\pm0.10}$ | 11.5% |
| MovieLens-1M | $15.94_{\pm0.20}$ | $18.24_{\pm0.27}$ | $14.12_{\pm0.17}$ | $15.80_{\pm0.21}$ | 11.9% | $15.47_{\pm0.11}$ | $20.51_{\pm0.18}$ | 32.6% | $18.23_{\pm0.13}$ | $20.43_{\pm0.25}$ | 12.1% |
| RECALL@10 – LOW-DEGREE USERS (LOWER PERCENTILE) | | | | | | | | | | | |
| Amazon-Book | $3.75_{\pm0.12}$ | $4.64_{\pm0.19}$ | $3.65_{\pm0.14}$ | $3.93_{\pm0.13}$ | 7.6% | $4.57_{\pm0.10}$ | $4.74_{\pm0.10}$ | 3.8% | $2.77_{\pm0.15}$ | $2.93_{\pm0.12}$ | 5.8% |
| Anime | $10.10_{\pm0.28}$ | $12.84_{\pm0.24}$ | $14.64_{\pm0.64}$ | $15.15_{\pm0.57}$ | 3.5% | $11.41_{\pm0.08}$ | $12.61_{\pm0.06}$ | 10.5% | $13.34_{\pm0.25}$ | $14.52_{\pm0.30}$ | 8.8% |
| Gowalla | $7.22_{\pm0.28}$ | $7.88_{\pm0.17}$ | $3.59_{\pm0.06}$ | $3.74_{\pm0.07}$ | 4.2% | $7.82_{\pm0.21}$ | $7.96_{\pm0.13}$ | 1.8% | $6.40_{\pm0.15}$ | $6.66_{\pm0.17}$ | 4.2% |
| Yelp-2018 | $3.45_{\pm0.12}$ | $3.64_{\pm0.13}$ | $2.44_{\pm0.06}$ | $2.57_{\pm0.10}$ | 5.5% | $3.60_{\pm0.12}$ | $3.73_{\pm0.16}$ | 3.6% | $2.62_{\pm0.16}$ | $2.92_{\pm0.19}$ | 11.5% |
| MovieLens-1M | $6.60_{\pm0.15}$ | $7.54_{\pm0.18}$ | $5.73_{\pm0.11}$ | $7.07_{\pm0.14}$ | 23.4% | $6.90_{\pm0.18}$ | $8.22_{\pm0.22}$ | 19.2% | $7.66_{\pm0.22}$ | $8.54_{\pm0.23}$ | 11.5% |
| RECALL@10 – OVERALL | | | | | | | | | | | |
| Amazon-Book | $3.62_{\pm0.22}$ | $4.46_{\pm0.17}$ | $3.81_{\pm0.17}$ | $3.99_{\pm0.08}$ | 4.8% | $4.44_{\pm0.06}$ | $4.56_{\pm0.07}$ | 2.6% | $2.81_{\pm0.24}$ | $3.00_{\pm0.26}$ | 6.7% |
| Anime | $11.12_{\pm0.18}$ | $12.86_{\pm0.22}$ | $13.45_{\pm0.26}$ | $13.71_{\pm0.26}$ | 1.9% | $11.43_{\pm0.08}$ | $12.71_{\pm0.04}$ | 11.2% | $14.19_{\pm0.46}$ | $14.64_{\pm0.39}$ | 3.2% |
| Gowalla | $7.43_{\pm0.06}$ | $7.70_{\pm0.12}$ | $4.01_{\pm0.07}$ | $4.04_{\pm0.08}$ | 0.8% | $7.57_{\pm0.18}$ | $7.72_{\pm0.09}$ | 2.0% | $6.56_{\pm0.17}$ | $6.77_{\pm0.21}$ | 3.4% |
| Yelp-2018 | $3.39_{\pm0.06}$ | $3.73_{\pm0.09}$ | $2.32_{\pm0.09}$ | $2.41_{\pm0.03}$ | 3.7% | $3.67_{\pm0.12}$ | $3.80_{\pm0.11}$ | 3.6% | $2.83_{\pm0.17}$ | $3.36_{\pm0.24}$ | 18.7% |
| MovieLens-1M | $7.00_{\pm0.13}$ | $7.66_{\pm0.20}$ | $6.20_{\pm0.23}$ | $7.02_{\pm0.25}$ | 13.2% | $7.68_{\pm0.12}$ | $8.33_{\pm0.16}$ | 8.4% | $7.93_{\pm0.20}$ | $8.75_{\pm0.22}$ | 10.3% |

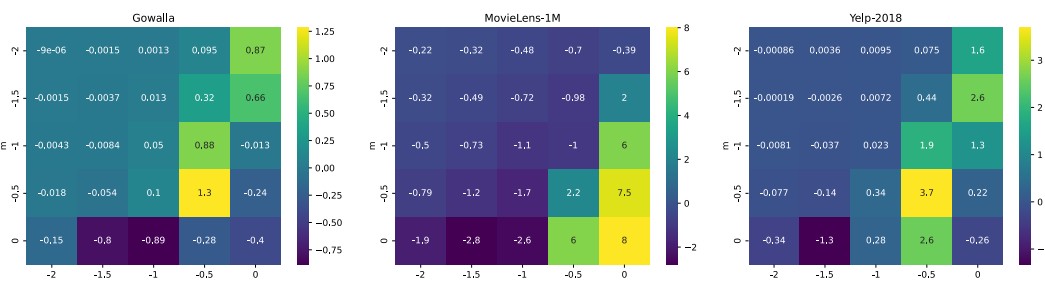

Figure 4: The sensitivity of TAG-CF to $m$ and $n$ in Equation (7). Numbers reported in these plots are performance improvement (%) brought by TAG-CF to MF trained by DirectAU [57] on Recall@20.

low-degree users. Because the contribution of the message passing over high-degree nodes has been mostly fulfilled by the inadvertent message passing during the training.

To quantitatively prove this line of theory, we incrementally up-sample low-degree training examples and observe the performance improvement that TAG-CF could introduce at each upsampling rate. If our line of theory is correct, then we should expect less performance improvement on low-degree users for a larger upsampling rate. The results are shown in Table 8. From this table, though upsampling low-degree users hurts the overall performance, we can observe that the performance improvement brought by TAG-CF for low-degree users decreases, as the upsampling rate increases.

According to this experiment, we can conclude that the more supervision signals a user receives (no matter for a low-degree or high-degree user), the less performance improvement message passing can bring. This experiment quantitatively shows why the performance improvement of high-degree users could be limited more than low-degree users. Because high-degree users naturally receive more training signals during the training whereas low-degree users receive fewer training signals.

# F Experiments on Ranking Metrics@10

This section shows the performance of all models as well as TAG-CF's improvement to them when evaluated with ranking metrics with 10 candidates. The results are shown in Table 9, with similar trends as we have observed in Table 2.

