# OpenReview forum: "How Does Message Passing Improve Collaborative Filtering?"
_NeurIPS.cc/2024/Conference — NeurIPS 2024 poster_

### Official Review · Reviewer_vJWR · 2024-07-08

**Soundness:** 3
**Presentation:** 3
**Contribution:** 3
**Rating:** 6
**Confidence:** 4

**Summary:**

This paper rethinks the application of message passing mechanisms in collaborative filtering methods and makes two key findings: 1) Message passing (MP) improves collaborative filtering primarily through the forward pass rather than the backward propagation process, and 2) MP is more effective for cold-start users compared to warm users. Based on these findings, this paper proposes a novel test-time aggregation method tailored for collaborative filtering, supported by theoretical derivations to demonstrate its validity. Finally, extensive experiments validate the effectiveness of the proposed method.

**Strengths:**

1. This paper rethinks the message passing mechanisms in collaborative filtering and makes two novel key findings.

2. The method has theoretical guarantees to ensure its effectiveness.

3. The structure of this work is logical, and the writing is well-organized.

**Weaknesses:**

1. Equation 4 seems unable to separate the forward pass neighbor representation from the backward pass gradient update in $s_{ij}$. Therefore, it is theoretically challenging to validate this finding, even though Table 1 empirically demonstrates that the forward pass primarily drives the performance improvement.

2. Typically, evaluation metrics for recommendation models use NDCG@10 and Recall@10, as users tend to focus on items ranked higher in the list while ignoring those ranked lower. Therefore, NDCG@10 and Recall@10 better reflect the true effectiveness of recommendation models.

3. It is necessary to explain some abbreviations, such as 'OOM' and '-'.

4. This paper has many grammatical problems, so it is suggested to improve it. For example, "TAG-CF is extremely versatile can be used as a plug-and-play module to enhance...".

**Questions:**

See weaknesses.

**Limitations:**

See weaknesses.

---

> ### Author Rebuttal · Authors · 2024-08-07
>
> Thank you for your valuable feedback. We really appreciate your acknowledgment of our proposal's significance, theoretical soundness, and presentation. Please see below for our detailed response:
>
> ## W1: Clarification on Equation 4.
> * The purpose of Equation 4 is comparing differences between a vanilla matrix factorization model and the LightGCN model in terms of how user-item similarities are computed. Once we find any differences between these two models, we can individually ablate/remove the additional inductive biases given by LightGCN and observe the performance downgrade to determine their significance to the recommendation performance. Specifically, for matrix factorization models, the similarity computation is extremely intuitive and it is as simple as the inner product between the corresponding user and item embeddings (i.e., $s_{ij}=\mathbf{u}_i^\intercal \cdot \mathbf{i}_j$). Whereas for LightGCN, the similarity computation is a combination of the same inner product and inner products between adjacent node embeddings. In this paper, we start our analysis from perspectives of forward and backward passes.
>
> * From the perspective of backward pass, the idea is that while incorporating similarities between adjacent nodes, the positive supervision signal from the target user-item pair is also back-propagated to adjacency node embeddings. This could improve the recommendation performance because of the idea behind collaborative filtering (i.e., in the case of matrix factorization, neighbors’ similarity scores are highly correlated with the similarity of the target user-item pair). To determine if this is the case and how much can such back propagation improve the recommendation performance, we remove the gradients flowed back to neighbor embeddings (i.e., in implementation, we freeze the detach gradient operation from pytorch) and accordingly observe the performance downgrade. This variant is denoted as LightGCN w/o grad., which surprisingly shows little performance downgrade.
>
> * For the perspective of forward pass, the idea is that to calculate the similarity between a pair of user and item, LightGCN also incorporates a weighted summation of similarities adjacent nodes. The underlying assumption is that neighbors’ similarity scores are highly correlated with the similarity of the target user-item pair. To determine if this operation is helpful, we need one ablation that only removes the numerical values of neighbors’ similarities and keeps everything else intact. In this case, the most straightforward way is removing the neighbor similarities from the calculation yet at the same time keeping the gradients brought by this calculation so that we do not ablate both backward and forward passes at the same time. And this is one of our variants coined LightGCN w/o neigh. Info, which proves that neighbor information drives the good performance of message passing in LightGCN.
>
> * This phenomenon indicates that (1) both additional representations passed from neighbors during the forward pass and accompanying gradient updates to neighbors during the back-propagation help the recommendation performance, and (2) within total performance gains brought by MP, gains from the forward pass dominate those brought by the back-propagation. Comparing LightGCN with LightGCN w/o neigh. Info, we notice that the incorporation of gradient updates brought by MP is relatively incremental (i.e., ~2\%). However, to facilitate these additional gradient updates for slightly better performance, LightGCN is required to conduct MP at each batch, which brings tremendous additional overheads. This rationale motivates our further investigation and the proposal of TAG-CF.
>
> ## W2: Additional resutsl for ranking metrics at 10.
> * Thanks for pointing this out. We explore this setting following existing works such as LightGCN and UltraGCN. We agree with you that Recall and NDCG at 10 can better reflect the true performance. Following your suggestion, we accordingly report Recall and NDCG at 10 in the one-page supplemental material and we observe that TAG-CF consistently brings performance improvement to baseline models when we explore more strict metrics, which further validates our proposal.
>
> ## W3, W4: Presentation improvement to the manuscript.
>
> * Thanks for your suggestions. In Table 2, “OOM” refers to out-of-memory and we use “-” to represent out-of-memory as well because we want to avoid repetitive denotations. We will clarify this point in the table caption. Furthermore, we will double check spelling and grammatical errors remaining in the manuscript to ensure the readability.
>
> **In light of our answers to your concerns, we hope you consider raising your score. If you have any more concerns, please do not hesitate to ask and we'll be happy to respond.**

---

> > ### Comment · Reviewer_vJWR · 2024-08-09
> >
> > Thank you for your detailed response; it has resolved my concerns. I will maintain my positive rating.

---

> > > ### Author Response · Authors · 2024-08-12
> > > **Thanks for your response.**
> > >
> > > We are delighted that our rebuttal satisfactorily leverages your concerns. We appreciate your kind responses and acknowledgement of our work.

---

### Official Review · Reviewer_3fxq · 2024-07-12

**Soundness:** 2
**Presentation:** 2
**Contribution:** 2
**Rating:** 5
**Confidence:** 5

**Summary:**

This paper investigates the role of message passing in collaborative filtering, providing empirical analysis of its impact. Based on their findings that message passing primarily benefits through additional neighbor representations during forward passes and helps low-degree nodes more than high-degree nodes, the authors propose TAG-CF, a test-time aggregation framework for collaborative filtering. TAG-CF demonstrates competitive performance with state-of-the-art graph-based collaborative filtering methods while significantly reducing computational overhead.

**Strengths:**

1. The paper experimentally analyzes the impact of message passing on collaborative filtering.
2. Based on the analysis results, an efficient TAG-CF method is proposed.
3. Extensive experiments were conducted on various datasets.

**Weaknesses:**

W1. Crucially, this paper overlooks related work, which do not require training.

W2.  The paper claims that TAG-CF can achieve performance similar to training-time message passing with just one aggregation at test time. More detailed analysis is needed to determine if this holds true in all situations and under what conditions these results occur.

W3. While it's claimed that performance improvement is greater for low-degree nodes, the theoretical explanation for this is somewhat lacking. A deeper analysis of why this phenomenon occurs is needed. In particular, it seems highly dependent on the degree cutoff.

**Questions:**

Q1. The authors have overlooked related work, such as GF-CF[1], BSPM[2], and SVD-AE[3], which are models without trainable parameters and training stages. The authors need to include these in their experimental results for comparison with test-time aggregation research or describe their commonalities and differences in the related work section. These should be mentioned as essential research before discussing test-time aggregation.

Q2. When the authors refer to low-degree nodes, is this based on the entire dataset or just the test or training set? If it's based on the entire dataset, couldn't the performance improvement be biased depending on how the test set is split?

Q3. The method for selecting the degree cutoff in TAG-CF+ seems somewhat heuristic. Is there a more systematic method for choosing this parameter?

Q4. What problems might arise when applying this to large-scale recommendation systems in practice?

Q5. Have you analyzed how TAG-CF's performance varies with the structural characteristics of the graph (e.g., node centrality, average path length, etc.)?

Q6. How does the dimension of user/item representations affect the performance of TAG-CF?

> [1] Shen, Yifei, et al. "How powerful is graph convolution for recommendation?." Proceedings of the 30th ACM international conference on information & knowledge management. 2021.
>
> [2] Choi, Jeongwhan, et al. "Blurring-sharpening process models for collaborative filtering." Proceedings of the 46th International ACM SIGIR Conference on Research and Development in Information Retrieval. 2023.
>
> [3] Hong, Seoyoung, et al. "SVD-AE: Simple Autoencoders for Collaborative Filtering." arXiv preprint arXiv:2405.04746 (2024).

---

> ### Author Rebuttal · Authors · 2024-08-07
>
> Thank you for your valuable feedback. We really appreciate your acknowledgment of our proposal's efficiency and our experiment's extensiveness. Please see below for our detailed response:
> ## W1, Q1: Missing related works
> * Thanks for pointing this out. Please see G1 (general response) for details. In summary, we (i) add experiments over 4 efficient baselines, and (ii) add a dedicated section about related efficient training for matrix factorization (MF). TAGCF consistently improves efficient MF methods (i.e., GC-FC, SVD-GCN, SVD-AE, and Turbo-CF). On average, TAGCF improves these methods by 5.6%, 8.9%, 3.9%, 7.8%, and 7.6% on our five datasets resp.
> ## W2: Analysis w.r.t. behaviors of TAG-CF
> * In our paper, we claim that test-time message passing (MP) improves the performance of MF methods similarly to training-time MP. We discover that MP (both at training and testing time) helps low-degree users more than high-degree ones. In Table 2, the majority of cases (15/18 cases, or 83.3%) support this finding. In rare cases where high-degree users benefit more, the gap between groups is marginal and our proposed TAGCF still improves the performance (i.e., 6.7%, 2.3%, and 1.8% overall NDCG improvement). TAGCF consistently improves the performance of MF methods over 6 datasets (Table 2). Besides, we also apply TAGCF to training-free baselines (i.e., GFCF, SVD-GCN, SVD-AE, and Turbo-CF), as you suggest, and we observe that TAGCF consistently improves upon them. The added experiments can be found in the 1pg supplement. On average, TAGCF improves 4 training-free methods by 5.6%, 8.9%, 3.9%, 7.8%, and 7.6% on five datasets resp.
> ## W3: Analysis and discussion about degree cut-off
> * We agree with you that the behaviors of TAGCF are dependent on the degree cutoff, as we show that MP (both at training and testing time) improves for low-degree users more than high-degree ones. Empirically, we observe that the majority of cases (15/18 in Table 2) support our finding. Moreover, we prove that both BPR and DirectAU optimize the objective of MP with some additional regularization (i.e., dissimilarity between non-existing user/item pairs for BPR, and representation uniformity for DirectAU).
> * Combining this theory with the prior empirical observations, we show that these two supervision signals  inadvertently conduct MP in the backward step, even without explicitly treating interaction data as graphs (see lines 219-237). To substantiate, we slowly upsample low-degree training users and observe the improvement that TAGCF introduces at each upsampling rate (see Appx. E).
> ## Q2: Definition of low-degree user
> * We agree with your point: when the full set is used to calculate degree, experiments might be biased towards testing splits. We use the _training set alone_ to calculate user degrees, as our findings are closely correlated with the amount of exposure a user embedding gets _during training_.
> ## Q3: Degree cut-off selection
> * There are 2 strategies to select degree cut-offs in TAG-CF+. One optimizes ranking performance, and the other focuses on budget (i.e. performance/cost ratio).
> * For the former: We treat degree cut-offs as hyper-parameter tuning and select the one that with best validation metrics. We explore this in the original draft (circled optimal points in Figure 2/3).
> * For the latter: GNN inference in industrial settings is prohibitively expensive especially for high-degree users, as their receptive field increases exponentially. As shown in Figure 2, performance gain from TAG-CF+ plateaus with a relatively small user degree. In these cases, the cost for further conducting TAG-CF+ on high-degree users grows exponentially, but doing so brings limited gains. Hence, we can also select the cut-off s.t. the performance/cost ratio starts to plateau. We will add this discussion to our manuscript.
> ## Q4: Challenges in large-scale RecSys
> * The challenges of applying TAGCF to large-scale RecSys are the utilization of graphs in large-scale ML pipelines. There are often O(100 million) users/items and O(trillion) of edges connecting them. Simply loading the entire sparse interaction matrix into CPU RAM is then infeasible. TAGCF utilizes graph knowledge while avoiding most of the computational overhead of MP. It is extremely flexible, simple to implement, and enjoys benefits of graph-based CF with minimal cost.
> ## Q5: Other graph characteristics
> * Thanks for pointing this out. We conduct additional experiments on Katz and Betweenness centrality. We split users equally into 3 buckets according to these characteristics and show the improvement brought by TAGCF to each bucket. Since these characteristics are expensive to compute, we only conduct this experiment on MovieLens-1M. We observe that improvement gradually increases as measurements go down, which echoes our observations on node degree.
>
> ||NDCG@20 Impr.|Recall@20 Impr.
> |:-|:-:|:-:|
> ||Katz
> High|29.2%|7.2%
> Med|33.4%|7.9%
> Low|35.2%|8.9%
> ||Betweenness
> High|27.8%|6.9%
> Med|34.7%|8.1%
> Low|35.6%|9.0%
> ## Q6: Embedding dimension
> * We conduct experiments on the improvement from TAGCF to MF with different dimensions. Due to time limitations and costs for training larger models, we only conduct experiments on Anime and MovieLens-1M. We observe that the performance of MF increases as the dimension increases. TAG-CF can consistently improve the performance of MF models with similar trends as observed in the original draft.
>
> |Dimension|64|128|256
> |:-|:-:|:-:|:-:|
> || Anime
> NDCG@20 MF|24.01|30.14|34.73
> +TAG-CF|27.25(9.8%)|32.82(9.0%)|37.96(9.3%)
> Recall@20 MF|29.15|35.78|38.43
> +TAG-CF|31.95(6.9%)|38.10(6.5%)|41.01(6.7%)
> ||MovieLens-1M
> NDCG@20 MF|22.51|24.34|25.08
> +TAG-CF|29.65(31.7%)|32.18(32.2%)|32.73(29.57)
> Recall@20 MF|26.30|28.82|29.57
> +TAG-CF|28.40(8.0%)|31.09(7.9%)|31.87(7.7%)
>
> **In light of our answers to your concerns, we hope you consider raising your score. If you have any more concerns, please do not hesitate to ask and we'll be happy to respond.**

---

> > ### Comment · Reviewer_3fxq · 2024-08-12
> >
> > Thank you for addressing my concerns. I raise my score to 5.

---

> > > ### Author Response · Authors · 2024-08-12
> > > **Thanks for your reply.**
> > >
> > > We greatly appreciate your valuable insights and constructive feedback on our paper. It's truly encouraging to see that our response effectively addresses the concerns you raised. Thank you for recognizing the efforts we've put into our work.

---

### Official Review · Reviewer_yNfp · 2024-07-16

**Soundness:** 3
**Presentation:** 3
**Contribution:** 3
**Rating:** 7
**Confidence:** 4

**Summary:**

The paper investigates on an interesting topic: how message passing is playing a role in graph-based recommender systems. Upon experiments on LightGCN model, authors posit the advantages of message passing lie in the forward pass to aggregate neighborhood information while at the same time, backward propagation has little effect on the results. Based on this assumption, authors propose test-time aggregation for collaborative filtering where the representation of users/items is aggregated at the testing time for more efficient training.

**Strengths:**

1. The paper identifies roles for message passing in graph-based recommendation.
2. It provides more insights for researchers to understand the mechanism behind graph-based recommendation.
3. The proposed method is efficient.

**Weaknesses:**

1. Lack of pre-training description
2. Lack of hyper-parameter analysis

**Questions:**

1. What is the detailed pre-training method for the method? I am assuming the pre-training strategy is impactful for the results.
2. An experiment on different pre-training methods is needed.
3. Authors design m,n hyper-parameters to adjust the neighborhood impact. Are there any findings we can obtain from the two hyper-parameters? I would like to see hyper-parameter testing on them.

**Limitations:**

See weakness

---

> ### Author Rebuttal · Authors · 2024-08-07
>
> Thank you for your valuable feedback. We really appreciate your acknowledgment of our proposal's significance, insights, and efficiency. Please see below for our detailed response:
>
> ## W1, Q1, Q2: Pre-training strategies for MF models
> * We agree with you that the pre-training strategies are impactful for the results, as TAG-CF is a test-time augmentation framework whose performance significantly depends on the starting performance of the base model that it improves on. In Table 2 of our original manuscript, we applied TAG-CF to MF methods pre-trained by ENMF, DirectAU, and UltraGCN. In Table 5, we also applied TAG-CF to MF derived by BPR. All these applications of TAG-CF to MF methods pre-trained by different methods show consistent performance improvement. Nevertheless, as suggested by other reviewers, we also apply TAG-CF to four other training-free matrix factorization methods (i.e., GFCF, SVD-GCN, SVD-AE, and Turbo-CF). As shown in the one-page additional material, TAGCF can still consistently improve the performance of these efficient matrix factorization methods (i.e., GC-FC, SVD-GCN, SVD-AE, and Turbo-CF). On average, TAGCF improves these four methods by 5.6%, 8.9%, 3.9%, 7.8%, and 7.6% on Amazon-book, Anime, Gowalla, Yelp-2018, and  MovieLens-1M respectively.
>
> ## W2, Q3: Sensitivity experiments for m and n.
> * Thanks for pointing this out. Please refer to G2 in our general response for details.
>
> **In light of our answers to your concerns, we hope you consider raising your score. If you have any more concerns, please do not hesitate to ask and we'll be happy to respond.**

---

> > ### Comment · Reviewer_yNfp · 2024-08-12
> > **keep score unchanged**
> >
> > I will keep my positive score unchanged. Thank you for author's rebuttal

---

> > > ### Author Response · Authors · 2024-08-12
> > > **Thanks for your reply.**
> > >
> > > Your valuable insights and constructive feedback on our paper are deeply appreciated. We are grateful for acknowledging the dedication we've invested in our research.

---

### Official Review · Reviewer_jYGx · 2024-07-31

**Soundness:** 2
**Presentation:** 3
**Contribution:** 3
**Rating:** 4
**Confidence:** 4

**Summary:**

This paper investigates the role of message passing (MP) in collaborative filtering (CF). Unlike most GNN-based CF research, which assumes that performance gains arise from improved representation learning through GNNs, this work questions that assumption. Through empirical experiments and theoretical analyses, the paper finds that the key contributions of MP in CF are: 1) forward message passing rather than back-propagation, and 2) benefits for low-degree users instead of high-degree users.

Based on these observations, this work propose a simple and efficient plug-in method called TAG-CF. TAG-CF can be easily integrated into any well-trained CF model by performing a single layer of message passing during the inference stage. To further reduce computational workload, TAG-CF+ conducts MP only on low-degree users. Experiments on four datasets verify the effectiveness of TAG-CF in improving CF models with almost no additional computational overhead.

**Strengths:**

1. This paper focuses on an important open problem: what is the role of MP plays in CF. This problem is worth investigating, and this work attempts to answer it from both empirical and theoretical perspectives.
2. This article is easy to follow, with a natural transition from the analysis of MP to the proposal of TAG-CF.
3. The proposed TAG-CF is efficient and effective. The plug-in nature of the method ensures the industrial application of the proposed method.

**Weaknesses:**

1. The discussion following Theorem 1 "both BPR and DirectAU optimize the objective of message passing" is not rigorous enough. Here are some aspects regarding to the discussion:
   * The assumption that $\|\mathbf{u}_i\|^2=\|\mathbf{i}_j\|^2=1$ does not hold in LightGCN, thus the discussion on the theorem is only empirical but not strictly hold for LightGCN. In other words, the BPR loss for real-world LightGCN may not be upper-bounded by the objective of message passing.
   * Even if the claim "both BPR and DirectAU optimize the objective of message passing" holds, there should be an additional experiment to show that directly train a CF model on the objective of message passing achieves comparable performance of LightGCN-BPR.

2. Important baselines such as GFCF [1] and SVD-GCN [2] are missing. These works also dedicated to speed up GNN-based CF. On Gowalla dataset, both GFCF and SVD-GCN cost only around 30s for training, which is more efficient than the proposed TAG-CF. Moreover, the reported Recall@20 on Gowalla are 0.1849 and 0.1905 respectively, also shows comparable or better performance. The authors need to add these baseline in the experiment and discuss what is the advantage of TAG-CF compared to these baselines.

3. According to Figure1, it is not sufficient to draw the conclusion that "Message Passing in CF Helps Low-degree Users More Compared with High-degrees". In Yelp dataset, the improvement of degree=16 users is only about 15%, which is less than the value of most higher-degree users (degree from 17-34). And what is the improvement for the users whose degree is less than 16? Still, this empirical experiment on only two dataset is not enough to support this conclusion.

[1] Yifei Shen, et al., “How Powerful is Graph Convolution for Recommendation?” CIKM 2021

[2] Shaowen Peng, et al., “SVD-GCN: A Simplified Graph Convolution Paradigm for Recommendation”. CIKM 2022

**Questions:**

Besides the weakness part, I also have some questions:

1. What is the setting of w/o both in Table 1. If it is a MF model using BPR, how can it achieve such good performance (Recall@20=18.42). In Table 2, MF model also shows superisingly good performance on most dataset, almost comparable to LightGCN, how does the MF set and tuned in the experiment?
2. In Table 2, low degree users' performance even better than overall performance, which seems to be inconsistent with Figure 1, are there any explanations?
3. How does m and n be tuned in the experiment? Are there any sensitivity experiments on these important hyper-parameters?

**Limitations:**

See weaknesses.

---

> ### Author Rebuttal · Authors · 2024-08-07
>
> Thank you for your valuable feedback. We really appreciate your acknowledgment of our paper’s significance, written quality, and practicality. Our detailed response to your concerns is as follows:
> ## W1.1: $|u_i|^2 = |i_j|^2 = 1$ does not hold in LightGCN
> * We agree that $|u_i|^2 = |i_j|^2 = 1$ does not hold in LightGCN due to the message passing (MP) and the mean averaging across embeddings from different MP layers. All of these operations will void this assumption. However, the main purpose of this theorem to support our claim that matrix factorization (MF) methods with trending objective functions (e.g., BPR and DirectAU) partially optimize the objective of MP with some additional regularization (i.e., dissimilarity between non-existing user/item pairs for BPR, and representation uniformity for DirectAU). Hence, directly optimizing these two objectives for MF partially fulfills the effects brought by MP during the back-propagation. We will refine our theorem and restrict it to MF methods, where the assumption of $|u_i|^2 = |i_j|^2 = 1$ makes sense. We sincerely appreciate your constructive feedback and the refined theorem will be: “During the training of MF methods, objectives of BPR and DirectAU are strictly upper-bounded by the objective of message passing.”
> ## W1.2: Directly training MF using the message passing objective
> * The objective of MP layer is $\mathcal{O}=\min_\mathbf{Z}\{tr(\mathbf{Z}^\intercal\mathbf{L}\mathbf{Z}))\}$, which enforces embeddings of adjacent node to be similar. Training one MF model directly optimizing this objective is equivalent to BPR without negative sampling or DirectAU without the uniformity term, which will lead to overfitting and collapsing issues.
> * The purpose of our theorem is to connect the gain brought by the MP to node degree. Our theorem shows that BPR and DirectAU partially conduct inadvertent MP during the back-propagation. In the case of CF, the amount of training signals for a user is directly proportional to the node degree. High-degree active users naturally benefit more from the inadvertent MP from objective functions, because they acquire more training signals. Hence, when explicit MP is applied to CF methods, the performance gain for high-degree users is less significant than that for low-degree users. Because the contribution of the MP over high-degree nodes has been mostly fulfilled by the inadvertent MP during the training.
> * Retrospecting on your suggestion, we believe that studying the direct correlation between the objective of MP and matrix factorization models can provide understanding towards the role of MP for CF. Hence, we provide the value of the MP objective quantified by $\sum_{(i,j)\in\mathcal{D}} ||\mathbf{u}_i - \mathbf{i}_j||^2$ at different training steps. The results for MF trained with DirectAU on MovieLens-1M is shown below:
>
> |Training steps (epoch)|0|2|4|6|8|10
> |:-|:-:|:-:|:-:|:-:|:-:|:-:|
> |$\|\|u_i-i_j\|\|^2$|0.931|0.493|0.478|0.464|0.461|0.461
>
> * We notice that the objective of MP is gradually optimized as the training progress and plateaus when the model converges, which echos with our finding that it optimizes the objective of MP with some additional regularizations.
> ## W2: Missing important baselines
> * Thanks for pointing this out. Please see G1 (general response) for details. In summary, we (i) add experiments over 4 efficient baselines, and (ii) add a dedicated section about related efficient training for matrix factorization (MF). TAGCF consistently improves efficient MF methods (i.e., GC-FC, SVD-GCN, SVD-AE, and Turbo-CF). On average, TAGCF improves these methods by 5.6%, 8.9%, 3.9%, 7.8%, and 7.6% on our five datasets resp.
> ## W3, Q2: Insufficient conclusion in Figure 1
> * In Figure 1, we indeed observe a local increasing trend in performance improvement from degree 16 to 20. However, when we compare the performance improvement of the whole low-degree users to that of the high-degree users, we can still observe that the improvement for low-degree users is larger than high-degree users (e.g., 4.8% on low-degree vs. 2.6% on overall for ENMF and 11.2% vs. 10.4% for UltraGCN). We do admit that there are certain cases where the low-degree improvement is slightly worse than the overall (i.e., 1.7% on low-degree vs. 1.8% on overall for MF). In fact, in these cases, the gaps in-between are very marginal (i.e., 0.1%) compared with gaps we observe for other cases (i.e., 2.2% and 0.8%). In Table 2, only 3 out of 18 cases (i.e., 16.7%) are scenarios where low-degree improvement is smaller and all remaining 15 cases (83.3%) support our findings that low-degree improvement is larger. Besides, even in these three rare cases, our proposed TAGCF still improves the performance (i.e., 6.7%, 2.3%, and 1.8% overall improvement in NDCG). So we believe that the conclusion we draw in this section is sufficient given abundant supporting evidence from experiments.
> ## Q1: Setting in Table 1
> We train both MF and LightGCN using the DirectAU loss and explore the same setting we have for Table 2. We individually tune hyper-parameters for MF and LightGCN (i.e., learning rate, batch size, and weight decay) to make sure that all models perform at their utmost, so that our findings are not based on weaker baselines. A vanilla MF model trained by DirectAU can already outperform LightGCN trained with BPR loss, which is the reason why we explore DirectAU loss for both MF and LightGCN in the first place. When we train both models using DirectAU, their performance gap is not as big as it is for gaps between models trained with BPR loss. We will include a dedicated section describing the hyper-parameters we used for each dataset.
> ## Q3: Sensitivity experiments for m and n
> * Thanks for pointing this out. Please refer to G2 in our general response for details.
>
> **In light of our answers to your concerns, we hope you consider raising your score. If you have any more concerns, please do not hesitate to ask and we'll be happy to respond.**

---

### Author Rebuttal · Authors · 2024-08-07

We thank the reviewers for their feedback and constructive suggestions. We are pleased that most reviewers appreciated **the promising efficiency and effectiveness of our proposal**, e.g.,: "the proposed TAG-CF is efficient and effective and ensures the industrial application of the proposed method (jYGx)", "the proposed method is efficient (yNfp)", and "efficient TAG-CF method is proposed (3fxq)". Besides, most reviewers also recognized **the promising impact of our research**, said: "focuses on an important open problem (jYGx)", "provides more insights for researchers (yNfp)", and "rethinks the message passing and makes two novel key findings (vJWR)".

**[G1: Additional experiments and discussions w.r.t. related works]** At the same time, multiple reviewers are concerned about the coverage of our related works and experiments, especially for those efficient baselines. We appreciate the insightful suggestions and agree that our work is relevant to other efficiency-focused efforts in recommender systems. To leverage this comment, we not only conduct additional experiments over four efficient baselines (i.e., GFCF [1], SVD-GCN [2], SVD-AE [3], and Turbo-CF [4]) but also add a dedicated section discussing related works about efficient training for matrix factorization. Before we dive into the experimental results, we would like to further note that TAG-CF is a test-time augmentation framework such that TAG-CF needs to be coupled with existing matrix factorization baselines (e.g., MF-BPR, MF-DirectAU, UltraGCN, GFCF, SVD-GCN, etc). It does not make sense to compare the performance of TAGCF across baselines because the better performance might come from TAGCF itself or a more performing backbone matrix factorization model, making the comparison between baselines non-trivial. In our original manuscript, we apply TAGCF to a series of matrix factorization methods (i.e., MF-BPR, ENMF, and UltraGCN) and TAGCF consistently enhances their performance. Following this setting, we also apply TAGCF to other matrix factorization methods as suggested by reviewers. Their performance and the improvement brought by TAGCF are systematically presented in the one-page pdf. From Table 10, we can observe that TAGCF can still consistently improve the performance of these efficient matrix factorization methods (i.e., GC-FC, SVD-GCN, SVD-AE, and Turbo-CF). On average, TAGCF improves these four methods by 5.6%, 8.9%, 3.9%, 7.8%, and 7.6% on Amazon-book, Anime, Gowalla, Yelp-2018, and MovieLens-1M respectively.

Besides additional experimental results, we also drafted a section discussing related works as follows:

Efficient efforts in marix factorization. A branch of research specifically focuses on improving the efficiency of matrix factorization [1,2,3,4,5]. For instance, GFCF [1] and Turbo-CF [4] explore graph signal processing to linearly convolve the interaction matrix and use the resulted matrix directly for recommendation without training. Furthermore, SVD-GCN [2] and SVD-AE [3] utilize a low rank version of the interaction matrix to further accelrate the convolution efficiency and yet remain the promising performance. Besides, BSPMs [5] studies using diffusion process to gradually reconsturct the interaction matrix and achieves promising performance with fast processing. In parallel with these existing efforts, we propose to enhance any existing matrix factorization method through test-time augmentation that harnesses graph-based heuristics.

[1] Shen, et al., How Powerful is Graph Convolution for Recommendation? CIKM21

[2] Peng, et al., SVD-GCN: A Simplified Graph Convolution Paradigm for Recommendation. CIKM21

[3] Hong, et al. SVD-AE: Simple Autoencoders for Collaborative Filtering. IJCAI24

[4] Park, et al. Turbo-CF: Matrix Decomposition-Free Graph Filtering for Fast Recommendation. SIGIR24

[5] Choi, et al. Blurring-sharpening process models for collaborative filtering. SIGIR23

**[G2: Additional hyper-parameter experiments on m and n]** Furthermore, multiple reviewer also ask about TAG-CF's sensitivity to hyper-parameters (i.e., m and n in the test-time message passing). To mitigate this concern, we plotted heat maps of the performance improvement brought by TAGCF to MF with different m and n configurations and show the results in the one-page pdf. From Figure 4, we can observe that m and n are important for the success of TAG-CF. Fortunately, across datasets, the optimal selection of m and n is pretty similar (e.g., m=n=-0.5 or m=n=0). The other solution to automatically tune m and n could be initialzing m and n to -0.5 (i.e., the value that generally works well across datasets) and conducting gradient descent on them using the training loss. But in this work we observe that manually tuning them on a small set of candidates can already deliver promising results.


**Besides this general response, we also leveraged individual feedbacks in the designated section. We hope we have satisfactorily answered your questions. If so, could you please consider increasing your rating? If you have remaining doubts or concerns, please let us know, and we will happily respond. Thank you!**

Best regards, \
TAG-CF authors

---

### Decision · Program_Chairs · 2024-09-25

**Decision:**

Accept (poster)

**Comment:**

This paper investigates the role of message passing (MP) in collaborative filtering (CF) and proposes a novel test-time aggregation method called TAG-CF.

Key Strengths:

- Novel insights: The paper provides new perspectives on how message passing functions in graph-based CF, challenging existing assumptions.
- Efficiency: TAG-CF demonstrates competitive performance with state-of-the-art graph-based CF methods while significantly reducing computational costs.
- Versatility: The method can be easily integrated into various existing CF models as a plug-and-play module.
- Theoretical foundation: The authors provide theoretical derivations to support the validity of their approach.
- Extensive experiments: The paper includes comprehensive evaluations across multiple datasets and baselines.

Key Weaknesses:

- Overlooked related work: Initially, the paper failed to compare against or discuss some relevant efficient CF methods that do not require training (e.g., GF-CF, BSPM, SVD-AE). However, the authors addressed this in their rebuttal by adding experiments and discussions on these methods.
- Limited analysis of conditions: More detailed analysis is needed to determine under what conditions the performance of TAG-CF is - comparable to training-time message passing.
- Theoretical explanation: The theoretical justification for why low-degree nodes benefit more from message passing could be strengthened.
- Heuristic parameter selection: The method for selecting the degree cutoff in TAG-CF+ seems somewhat ad-hoc and could benefit from a more systematic approach.
- Evaluation metrics: Initially, the paper used NDCG@20 and Recall@20, while NDCG@10 and Recall@10 might better reflect real-world recommendation scenarios. The authors addressed this in their rebuttal by providing additional results for these metrics.

The paper presents a novel and efficient approach to collaborative filtering that challenges existing assumptions about message passing in graph-based recommender systems. The proposed TAG-CF method shows promise in terms of both performance and computational efficiency. While there were some initial oversights in related work comparisons and evaluation metrics, the authors have addressed most of these concerns in their rebuttal and reviewers increased their ratings. The paper is likely to generate interest and discussion in the recommender systems community.

While the paper is very close to be rejected, it improved significantly during the review process and the question of integration of other signal in collaborative filtering is both important and difficult from an industrial perspective. I feel the idea of MP is worth the be presented and discussed at the conference.  Also one of the reviewers didn't engaged into discussion yet other reviewers with similar concerns upgraded their scores after the rebuttal.

Please include Tabla 1 better in the final version.